# Low-Carbon Practices and Cultural Adaptation Among Older Chinese Migrants: Insights from Walking Interviews on Environmental Policy and Social Integration

**DOI:** 10.3390/ijerph22060832

**Published:** 2025-05-25

**Authors:** Qing Ni, Hua Dong, Antonios Kaniadakis

**Affiliations:** 1Brunel Design School, College of Engineering, Design and Physical Sciences, Brunel University of London, Uxbridge UB8 3PH, UK; qingni@brunel.ac.uk; 2Department of Computer Science, College of Engineering, Design and Physical Sciences, Brunel University of London, Uxbridge UB8 3PH, UK; antonios.kaniadakis@brunel.ac.uk

**Keywords:** older Chinese migrants, low-carbon behaviors, cultural adaptation, walking interviews, frugality, environmental policy, social integration

## Abstract

This study employs walking interviews to examine the low-carbon practices, cultural adaptation, and policy awareness of older Chinese migrants in the UK within their everyday environments. A total of 20 participants were interviewed in public spaces such as parks, supermarkets, and their homes. Using contextual thematic analysis, the study identifies key factors influencing their environmental behaviors. The findings reveal the following: (1) Language barriers, economic pressures, and social isolation limit migrants’ understanding of environmental policies. Many participants rely on self-sufficient ethnic community networks rather than engaging with mainstream sources; (2) Generational differences are evident—younger migrants demonstrate greater theoretical awareness of environmental policies, whereas older migrants exhibit stronger low-carbon behaviors through energy conservation and waste reduction; (3) A balance between cultural identity and consumption habits—while some migrants adjust their dietary, spending, and linguistic habits, core cultural values such as frugality and family responsibility remain unchanged. This study highlights the value of walking interviews in capturing situational insights into low-carbon behaviors and cultural adaptation. It provides empirical evidence for government agencies and community organizations, advocating for cross-cultural environmental education and improved policy communication. Recommendations include targeted environmental training, community-based volunteer initiatives, intergenerational environmental education, and policy dissemination through WeChat, Chinese communities, and ethnic networks. These measures can help bridge the generational gap in policy awareness and promote social integration among older Chinese migrants.

## 1. Introduction

Climate change is a global challenge that requires collective action in all sectors of society [1]. Recent estimates suggest that demand-side mitigation strategies, including changes in infrastructure, technology, and individual behavior, could reduce global greenhouse gas emissions by 40 to 70% by 2050 compared to current policy projections [2]. Although considerable attention has been paid to technological innovation and national policies, the role of individuals, particularly those of underrepresented communities, in climate action remains underexplored [3].

In aging societies like the UK, older adults are becoming increasingly important in climate discourses. Older populations may be concerned about climate change due to their vulnerability to extreme weather, a sense of intergenerational responsibility, and accumulated life experiences that shape their political and social attitudes [4]. Among them, older adults with migration backgrounds represent a growing demographic group [5,6], whose relationships with communities are shaped by aging, displacement, and social policy contexts [7,8].

Migrant communities are often absent from mainstream sustainability discourses [9,10]. This omission is problematic not only in terms of equity, but also because migrants may possess cultural practices that support sustainable living [11]. In particular, older Chinese migrants bring cultural values of thrift, moderation, and resourcefulness rooted in Confucian teachings and historical scarcity [12]. These values have continued to influence their behaviors in host countries, even as their formal engagement with environmental programs remains low.

Although many older Chinese migrants engage in low-carbon behaviors, they may not identify them as “environmental” [13,14]. Language barriers, unfamiliarity with UK systems, and limited social integration often exclude them from formal climate initiatives [15,16]. These factors, combined with intergenerational differences in climate awareness, create a complex landscape for them to understand their role in sustainability transitions [17,18].

This study focuses on older Chinese migrants in the UK—a group underrepresented in both environmental and migration studies. It asks the following questions:How are low-carbon behaviors among older Chinese migrants in the UK shaped by cultural memory, migration experience, and everyday adaptation to their new environment?How do older Chinese migrants perceive generational differences in sustainability attitudes and behaviors within their families or communities?How do social networks and migrant-specific contexts influence their access to low-carbon information?

In this study, we draw on the Value-Belief-Norm (VBN) theory [19] to explore how moral values influence everyday environmental behavior. At the same time, we incorporate ideas from Practice Theory [20,21] and Cultural Capital Theory [22,23] to better understand how older migrants’ habits are shaped by experience, routine, and social history. These perspectives shift the focus from individual choices to the wider social and cultural context in which behaviors take place.

From this view, frugality is not a conscious response to climate change campaigns, but a long-standing way of living rooted in Confucian ethics, past scarcity, and the habit of careful spending. For many older Chinese migrants, low-carbon actions arise not from environmental beliefs, but from values and habits passed down over time. By foregrounding the lived experiences of culturally marginalized groups, this study broadens the scope of sustainability discourse and calls for more inclusive, context-sensitive frameworks.

## 2. Research Background

### 2.1. Acculturatio

Understanding how older migrants adapt and act sustainably requires engaging with two key bodies of literature: acculturation theory [24] and environmental behavior models.

Berry’s acculturation framework identifies four adaptation strategies based on the degree to which individuals retain their heritage culture and adopt the host culture: assimilation, separation, integration, and marginalization [25]. This model is particularly relevant to older migrants, who often maintain strong cultural identities in the private sphere while adopting selective public behaviors for practical reasons [26].

Kim’s integrative theory of cross-cultural adaptation adds a temporal and communicative dimension. Adaptation is seen as a stress–adaptation–growth process, where communication with the host society plays a central role [27]. Older migrants, who may arrive later in life and face greater linguistic and systemic barriers, often remain in early-stage adaptation or selective acculturation [28,29,30]. These frameworks are particularly useful in interpreting the behavioral patterns of older Chinese migrants, who often display partial acculturation: maintaining traditional values such as thrift while adapting selectively to host country systems [31].

To explain environmental behavior, traditional models such as the Theory of Planned Behavior (TPB) emphasize individual attitudes, subjective norms, and perceived behavioral control [32]. However, TPB has been critiqued for underestimating the role of habitual, cultural, and contextual factors in shaping sustainable practices [33,34].

VBN theory (Stern, 2000) provides a framework to understand how environmental behaviors may arise from personal values, ecological worldviews, and feelings of moral obligation [19,35]. In our study, this theory helps frame how older migrants’ environmental behaviors might be rooted in long-standing values such as thrift, responsibility, and respect for nature—elements often transmitted across generations. However, in migration contexts where behavior is constrained by economic pressures or cultural displacement, VBN’s assumption of a linear progression from values to norms to behaviors may not fully apply. This insight motivates our complementary use of Practice Theory [20,21] and Cultural Capital Theory [22,23], which better captures the routinized, non-reflective, and context-dependent nature of many sustainable practices observed among our participants.

### 2.2. Frugality as Cultural Capital


Frugality among older Chinese migrants is not merely an individual habit—it functions as a form of cultural capital embedded in moral tradition, historical hardship, and collective identity [36,37]. In Chinese society, thrift is long linked to Confucian virtues such as moderation and responsibility [31,38]. The idiom “diligence and thrift” reflects a durable moral ideal that continues to shape practices across generations. Empirical studies have found that Chinese immigrants often display energy-saving habits distinct from native-born Western populations, including minimizing appliance use, reusing materials, and reducing food waste [18,38,39,40]. These behaviors stem not from environmental ideology but from ingrained cultural values.

Social reinforcement also plays a role: wastefulness is often stigmatized in Chinese communities, and maintaining thrift is seen as a matter of dignity or “face” [41]. Some migrants even express skepticism toward Western green consumption practices, which they perceive as costly or superficial [42,43]. Importantly, frugality may fade over time if purely economically motivated. As migrants achieve financial security, sustainable habits may weaken unless they remain tied to deeper cultural or moral values [44,45]

### 2.3. Cultural Memory, Scarcity, and Sustainable Practices


Frugality among older Chinese migrants reflects not only Confucian values but also lived experiences of material scarcity, particularly among those born before the 1960s who endured famine and hardship [18,40].

Frugality is also economically rational: many migrants manage limited financial resources while facing higher living costs in host countries. The existing literature has examined several factors contributing to the high savings rate among Chinese households. Studies have shown that precautionary savings for healthcare and old-age security play a significant role [46]. Additionally, expenditures on children’s education and housing impose considerable financial pressure [47,48,49]. Furthermore, research indicates that households with children tend to save more in preparation for future costs such as marriage and home purchases [50].

### 2.4. Intergenerational Differences in Environmental Behavior


As highlighted by Handy et al. in 2023 [51], research on the intergenerational transmission (IGT) of environmental behaviors is often grounded in socialization theory. This framework suggests that affiliations such as family or religious communities play a critical role in shaping individual behaviors by embedding shared values [52]. Within the household, parents model environmental practices—such as recycling, conserving energy, and minimizing waste—that children absorb through both explicit instruction and everyday observation [53]. These practices are internalized over time through embodied routines and unconscious imitation, as Bourdieu’s notion of habitus explains [54,55].

Environmental norms thus pass across generations not only through spoken values but through lived, habitual actions, with motivations shaped by the socio-economic and historical context in which each generation came of age [56,57,58]. In China, younger generations raised in the era of economic liberalization and globalization are typically more attuned to the environmental discourse but often lead more consumption-oriented lifestyles [59].

This generational divergence is further reinforced by differences in education and media exposure. Older migrants may continue frugal behaviors such as reusing materials or minimizing utility use, driven by long-standing habits or economic sensibilities. Meanwhile, their children may adopt similar practices—such as recycling or buying green products—based on formal environmental education or digital narratives around planetary responsibility. Despite outward similarities in behavior, the underlying motivations can differ markedly between generations [51].

Empirical research also suggests that younger Chinese migrants express higher levels of concern for climate change and are more likely to support environmentally friendly products. Yet they also tend to favor convenience and fast fashion, exposing a notable gap between their stated values and everyday consumption patterns [18].

These intergenerational differences may result in both friction and mutual learning within migrant families. While younger members might advocate for energy-efficient technologies or eco-conscious habits, older parents often embody sustainability through lived, habitual frugality—a practice rooted less in environmental ideology than in deeply held cultural norms [60].

### 2.5. Migrants’ Sustainability Practices in Context


Migrant environmental behaviors do not occur in a vacuum—they are shaped by both past experiences in the country of origin and the socio-political context of the host country [61,62]. In China, concepts like circular economy and ecological civilization have gained prominence in recent years [63]. However, older migrants who left before these discourses emerged often draw on practical norms rooted in necessity—such as mending broken items or never wasting rice.

In the UK, sustainability is widely institutionalized, with local councils running recycling programs, insulation grants, and public climate campaigns [64,65]. However, for older migrants with limited English and unfamiliarity with bureaucratic processes, these systems can feel alienating. Research indicates that older migrants often remain disconnected from mainstream sustainability schemes and instead rely on ethnic community networks or personal strategies (e.g., lowering the thermostat to save on bills) [10,39].

This disconnection reflects a broader issue: sustainability policy and migration governance often operate in silos [66]. Migrant voices are seldom included in environmental planning, leading to a sense that certain programs “are not for them.” [9,67,68]. Martínez-Alier’s distinction between the “environmentalism of the rich” and “environmentalism of the poor” captures this divide: the former is driven by ecological concern and often requires financial investment (e.g., installing solar panels), while the latter is rooted in thrift, necessity, and social justice [69]. For many older Chinese migrants, sustainability may be seen as a byproduct of saving money rather than a moral or political stance [10,70]. This perspective, while practical, challenges dominant framings of environmental citizenship [71]. At the same time, it opens up alternative pathways for low-carbon living rooted in cultural wisdom rather than technological solutions.

Recent research has also shown that migrants carry forward sustainability practices shaped before migration. These practices—such as active recycling, energy conservation, or reliance on public transport—are often established in contexts of resource scarcity and continue to guide behaviors in the host country [10,72,73]. For example, Head et al. found that ethnic minority migrants in high-income countries tend to maintain water-saving habits and frequent public transportation use, even after relocation.

Crucially, many of these actions are “quietly sustainable”—that is, they are neither highly visible nor aligned with dominant environmental discourses. Yet they still contribute meaningfully to sustainability goals. This aligns with the concept of “actually existing sustainabilities” [74], which emphasizes everyday practices not always labeled as sustainable but that effectively support environmental outcomes.

Evidence from global studies further reinforces these findings. In Australia, migrants expressed a desire to continue small-scale agriculture and culturally embedded gardening practices from their countries of origin [10], In Ghana, Abu et al. [73] showed that sustainable behaviors developed pre-migration—such as waste minimization and transport choices—were strong predictors of post-migration behaviors. Meanwhile, in Brussels, Zickgraf et al. [75] documented how migrants actively pressured local authorities to improve infrastructure, such as building more bike lanes and creating community gardens—motivated by comparisons with their places of origin.

Ultimately, if cities are to build inclusive sustainable futures, they must empower migrant voices and recognize these alternative sustainability pathways. For instance, in the USA, the city of Worcester’s 2021 Environmental Justice Policy states that “all communities—regardless of race, color, national origin, income, or language proficiency—must have a strong voice in environmental decision-making”. This policy serves as a model for embedding equity into urban sustainability governance [76].

## 3. Methodology

### 3.1. Research Design and Rationale

This study employed a qualitative approach using walking interviews to explore how older Chinese migrants in the UK understand and engage in low-carbon practices within their everyday lives [77,78]. Compared to seated interviews, walking interviews provide stronger contextual grounding and interactive potential, particularly suitable for uncovering the lived motivations, cultural routines, and spatial memories of older adults.

During an earlier exploratory phase, seated interviews and surveys were conducted with a broader cohort of older Chinese migrants to collect general perceptions on sustainability, climate policy, and cultural adaptation. While these dialogues offered valuable thematic insights, they also exposed limitations in capturing the deeper, embodied motivations behind behavior. Building on this, walking interviews were adopted to access more situated narratives and better understand the intersection between thrift, cultural identity, and environmental action.

Why walking interviews?

Walking interviews allow participants to anchor their reflections in familiar spaces. As they moved through gardens, kitchens, or neighborhood stores, participants pointed to specific practices—such as vegetable growing, reusing containers, or hunting for discounted items—embedding abstract narratives into tangible routines.

In practice, some participants were initially reserved during seated interviews due to cultural or migratory sensitivities. However, when placed in familiar environments, they became more relaxed and willing to share. For instance, one participant proudly showcased vegetables grown in their backyard; another described annual foraging traditions with younger migrants; yet another demonstrated how household waste was repurposed into useful items. Such insights might not surface in static interview contexts.

Thus, walking interviews provided rich, contextualized data and deepened our understanding of the spatial, cultural, and emotional dimensions of low-carbon behavior—highlighting the method’s unique value in migration and sustainability research.

Although this study was inspired by earlier phases of data collection, it constitutes a standalone investigation with a distinct methodological focus. Rather than statistically comparing migrant and native-born behaviors, the study seeks to interpret the culturally embedded practices of older migrants, emphasizing lived experiences and contextually grounded sustainability, not behavioral quantification.

### 3.2. Participant Recruitment and Sampling Strategy

This study involved 20 older Chinese migrants aged between 55 and 85, primarily residing in Greater London. Participants were recruited from boroughs with large Chinese populations—such as Kingston, Barnet, Hackney, and Islington—using a combination of offline and online outreach. Offline recruitment was conducted at local community hubs (e.g., the Kingston Chinese Community Centre and Chinese churches), where we received permission to distribute recruitment posters. Online outreach relied on snowball sampling and posts shared within Chinese-speaking Facebook groups and WeChat networks. Interested participants contacted the first author to arrange a walking interview.

Eligibility criteria focused on individuals with diverse migration histories whose lived experiences aligned with the study’s focus on low-carbon behavior, cultural values, and adaptation. These included marriage migrants, elderly dependents (e.g., grandparents relocated for family care), former labor migrants, and international students-turned-residents. In selecting the participants, we prioritized those who demonstrated reflective depth in prior interviews and represented variation in household structures, lifestyle routines, and sustainability engagement.

To better understand the diversity of observed behaviors, we adopted a purposive sampling approach along two key dimensions:Migration Background: Participants represented a range of routes to the UK, including educational, employment, and family-based migration.

Sustainability Orientation: Participants demonstrated varying degrees of environmental involvement—from highly active "environmental pioneers" to individuals facing barriers to engagement.

Although the sample may appear small, it aligns with the goals of qualitative research, which prioritizes depth, nuance, and contextual insight over statistical representativeness. The prior literature suggests that data saturation in qualitative interviews is typically achieved with 12–20 participants, particularly when the focus is on thematic exploration rather than population-level generalization [79,80]. Moreover, this study’s exploratory nature—centered on complex cultural meanings and lived practices—justifies the selected sample size for an in-depth, localized understanding. Future research may expand to other UK regions or comparative groups to enhance generalizability.

Finally, during the coding and analysis process, participants were informally grouped into three behavioral types to contextualize their sustainability motivations (summarized in Table 1):Environmental Pioneers—actively engaged in sustainability efforts beyond individual practices;Frugal Practitioners—whose behaviors stem from thrift and necessity rather than ecological ideology;Adaptive Integrators—who blend host country norms with inherited cultural values.

As shown in Table 1, the participants represented a diverse cross-section of older Chinese migrants, allowing comparisons between migration histories, behavioral patterns, and levels of social integration.

### 3.3. Research Instrument Design and Validation

The semi-structured interview guide was developed by the first author. The questions were informed by themes identified in the literature review (e.g., frugality, cultural adaptation, and low-carbon practices) and grounded in theoretical frameworks such as Practice Theory and the Value-Belief-Norm (VBN) model.

To ensure relevance and comprehensibility for the target population—Chinese migrants aged 55 and over—the guide was written in both English and Mandarin. It was pilot-tested with three members from a local Chinese older adult group. Their feedback informed revisions to the phrasing, sequencing, and length of the questions.

Two pilot walking interviews were conducted to refine the methodology:In the first pilot, a participant led the researcher on an extended outdoor walk, which proved logistically and physically challenging. As a result, the protocol was adjusted to prioritize shorter, more accessible walking routes.In the second pilot, a participant chose a crowded supermarket as the location, where ambient noise disrupted audio recording. Subsequent interviews were therefore conducted in quieter public settings or seated areas conducive to reflective conversation.

These pilot interviews informed the final methodological framework, enhancing participant’s comfort and safety. The interview questions were open-ended and designed to prompt reflections on daily routines, past experiences, environmental values, and community engagement.

Sample questions included the following:“Can you describe how you do your shopping in the UK? How is it different from how you used to shop in China?”“Have you taken specific steps to reduce waste or conserve energy? What motivates you to do this?”“Have you participated in any government or community programs related to the environment?”

The walking interview method functioned as both a spatial and narrative tool, enabling the researcher to observe and discuss behaviors in situ. As the participant walked through familiar environments, movement and contextual cues shaped the flow and depth of the conversation.

To ensure academic rigor and cultural sensitivity, the two co-authors reviewed and supervised the interview plan as experts.

### 3.4. Data Collection Procedure

Before each interview, participants received a detailed information sheet explaining the study’s purpose, ethical considerations, and procedures. Informed consent was obtained in writing, and all participants were made aware of their right to withdraw at any stage.

Walking interviews were held in locations of the participants’ choice, usually within their everyday environments such as local parks, markets, or residential areas. In some cases, the interview included brief walks followed by seated conversations in cafés or homes. The duration ranged from 30 min to nearly two hours, depending on participants’ preferences and mobility.

Interview questions were semi-structured and open-ended, designed to elicit participants’ reflections on themes such as the following:Everyday low-carbon behaviors;Frugality and consumption;Cultural adaptation and community life;Experiences with UK climate policies or environmental programs.

The flexible format allowed participants to guide the conversation toward aspects of daily life most meaningful to them. The interviews were conducted in Mandarin, Cantonese or English, according to each participant’s preference, and recorded using a portable audio recorder.

### 3.5. Data Analysis

All walking interviews were transcribed verbatim and translated into English where necessary. The transcripts were analyzed using thematic analysis [81], which allowed for the identification of recurring patterns related to environmental behavior, cultural values, and adaptation experiences.

The analytical process followed a structured approach:Familiarization: The researcher read through the full transcripts multiple times, making preliminary notes on key expressions and observations shared during the walks.Initial Coding: Using NVivo, open coding was applied to highlight specific statements, metaphors, or practices related to sustainability, migration, frugality, and identity. This phase generated an initial set of 18 codes.Theme Development: Codes with overlapping or conceptually similar meanings were clustered. For example, ‘language difficulties’ and ‘lack of access to public information’ were grouped under ‘Information Barriers’, while ‘use of second-hand items’ and ‘creative reuse’ were grouped under ‘Practical Frugality’.Theme Refinement: Themes were refined through iterative cross-coding and grouped based on semantic similarity and the frequency of co-occurrence, allowing the formation of coherent thematic clusters.Subthematic Structuring: For complex themes such as economic attitudes, three tiers of subthemes were introduced to reflect the nuances in participant responses. For example, under the theme of ‘Consumption and financial practices’, the subthemes included the following:“Responses to rising living costs”;“Frugality vs. aspirational consumption”;“Perceptions of waste and reuse”.

This multilevel coding structure allowed for both depth and comparability across participant experiences. The coding was discussed at meetings with the co-authors. The thematic framework was then mapped back onto the research questions and theoretical lenses discussed in the literature review.

Notably, the walking interview format enriched the data by contextualizing participants’ narratives in space. For example, references to energy-saving were often made while showing physical modifications in the home (e.g., insulated curtains or reused plastic bags), enhancing the embodied understanding of sustainability practices.

### 3.6. Trustworthiness and Triangulation

To ensure methodological rigour, this study employed multiple strategies aligned with qualitative research standards [82]:Credibility

Member validation was carried out by sharing the initial findings with a small group of participants and local Chinese community leaders. Their feedback was used to refine interpretations and ensure cultural accuracy.Prolonged engagement was maintained through multiple interactions with several participants, allowing for deeper trust and clarification.

Transferability

While the study focused on older Chinese migrants in urban UK contexts, detailed descriptions of participant profiles, cultural backgrounds, and environmental behaviors were provided to support transferability to other migrant populations or aging communities.

Dependability and Confirmability

A reflexive journal was kept throughout the research process to record methodological decisions, positionality reflections, and potential biases. These memos enhanced transparency in how themes were identified and interpreted.An audit trail of data coding, analysis steps, and software output (e.g., NVivo codebooks) was preserved for verification.

Methodological Triangulation

Although the walking interviews served as the primary data source, they were informed and complemented by insights gained from the earlier sit-down interviews. This allowed for the cross-verification of key themes, especially regarding frugality, language barriers, and social integration.

Walking interviews also enabled observational triangulation. For instance, participants’ self-described practices (e.g., “I always bring my own bag”) were often corroborated by visible evidence during the walk (e.g., showing hand-stitched fabric bags made from old clothing). This situated triangulation added ecological validity to the data.

## 4. Results

This chapter presents the key findings from the walking interviews, which offer rich, situated accounts of older Chinese migrants’ daily experiences and sustainability practices. Building on the thematic structure introduced in the methodology, the analysis is organized around four core areas that directly correspond to the study’s guiding research questions:Section 4.1 addresses Research Question 1, examining the barriers that prevent older Chinese migrants from engaging with local communities and environmental policies.Section 4.2 responds to Research Question 2, exploring the generational divergence between environmental awareness and actual low-carbon behaviors.Section 4.3 delves into Research Question 3, unpacking the historical, cultural, and economic roots of frugality as a form of sustainable practice.Section 4.4 extends the analysis by examining the dynamic interplay between adaptation and continuity in post-migration life, highlighting why some lifestyle habits shift while others persist.

To support the structure of this chapter, a visual summary of the eight major themes and their subthemes identified through thematic analysis of the walking interview data is presented in Appendix A. This analytical framework guided the organization of findings and facilitated the interpretation of older Chinese migrants’ lived experiences, particularly regarding cultural adaptation, frugality, and sustainability practices.

### 4.1. Integration and Policy Engagement:Barriers to Local Participation Among Older Chinese Migrants

This section responds to Research Question 1, examining why some older Chinese migrants struggle to integrate into local communities and engage with UK policy initiatives. Findings from the walking interviews reveal that multiple structural and cultural factors shape these challenges, including language barriers, cultural norms, economic pressures, and reliance on self-contained ethnic networks.

#### 4.1.1. Language Barriers: The First Hurdle to Integration

Limited English proficiency was consistently cited as one of the most significant obstacles to participation in UK society. Many older Chinese migrants relied primarily on Chinese-language media and social networks, which limited their exposure to local policies and services. As a result, participants described feelings of isolation, anxiety, and disempowerment in their early years of migration.

*“When I first arrived in the UK, I didn’t even know how to use the bus or read the coins. Life felt completely disorienting—like being a newborn, except filled with fear.”* — Shun (Female, 55+, Mainland China)

*“At university, I studied Russian, not English. When I came to the UK, I was too afraid to ask for directions. Now I can handle daily conversations, but I still struggle with formal language—legal documents, for example, are a nightmare. But I keep learning, even now at 79.”* — Rui (Female, 75+, Mainland China)

As shown in Figure 1, Rui often used English environmental learning materials during self-directed language study.

*“Many migrants make the mistake of staying within their language bubble. But that means you never really know what’s happening in the country.”* — He (Female, 60+, Malaysia)

#### 4.1.2. Cultural Differences and Social Barriers

##### Differences in Social Interaction Patterns

Western societies often emphasize individualism and independence in social life, while many Chinese migrants are used to family-centered and network-based relationships. This contrast can make participation in broader local communities feel unnatural or unimportant.

*“Most of my weekdays revolve around my children and grandchildren. Now that they’re older, I finally have time to attend weekend dance classes at the Chinese community center.”* — Fa (Female, 65+, Mainland China)

##### Discrimination and Prejudice

Several participants reflected on past experiences of racial or gender-based discrimination, which shaped their views on integration and social trust.

*“As a Chinese woman, I faced both sexism and racism in my early career, but I pushed through. You must keep improving yourself if you want to succeed in the West.”* — He (Female, 60+, Malaysia)

*“We came in the 1980s, and discrimination was everywhere. We tried to hide our ‘Chinese-ness’ just to blend in. Things have changed now—but I can’t tell if that’s because of our age or because China’s status has risen.”* — Jiao (Female, 60+, Mainland China)

*“Back home, we were the ‘pillar generation’. Here, we’re invisible.”* — Niqiu (Male, 55+, Mainland China)

These experiences of discrimination, while not directly linked to environmental motivations, often resulted in a sense of social detachment from mainstream British society. In response, many older migrants turned inward—relying more heavily on ethnic networks and traditional cultural values, including thrift and self-reliance.

#### 4.1.3. Political Apathy and Scepticism

##### Political Apathy

Some participants, especially from mainland China and Malaysia, admitted to a general indifference to local politics, shaped by past political environments where civic engagement was either discouraged or inaccessible.

*“If something doesn’t directly affect my life, I don’t pay attention to it.”* — Chen (Male, 55+, Malaysia)

*“I understand the importance of sustainability. But not everyone listens—especially Chinese people. Many just don’t care.”* — Shun (Female, 55+, Mainland China)

##### Scepticism Towards Local Policies

Others expressed cynicism toward government motives behind environmental and social policies, viewing them as revenue-driven rather than genuinely sustainable.

*“The government keeps changing, but meaningful environmental policies? Rare. I don’t really trust what politicians say.”* — Huang (Female, 60+, Taiwan)

*“A lot of these so-called ‘green policies’ are just hidden tax schemes. They create the problems, then sell the solutions.”* — Huang (Male, 70+, Hong Kong)

#### 4.1.4. Personality Traits and Self-Perception

Several participants attributed their reluctance to integrate to self-consciousness or fear of embarrassment in language use. This self-perception reinforced avoidance, creating a feedback loop of limited engagement.

*“Chinese and Taiwanese people tend to be shy. We fear making grammar mistakes. But why? They can’t speak our language either. We should be confident.”* — Huang (Female, 60+, Taiwan)

*“Many migrants isolate themselves in the Chinese circle. Their English doesn’t improve, so they become even more reluctant to speak. It’s a mindset issue.”* — Tao (Female, 55+, Mainland China)

#### 4.1.5. Economic Pressures and Ethnic Self-Sufficiency: Survival First, Integration Later

For many older Chinese migrants, economic survival and family responsibility consistently took precedence over community involvement or political engagement. Migration was often viewed pragmatically—as a way to support families in China, not necessarily as a path toward integration or civic participation in the UK.

*“Many Chinese in the takeaway industry came as undocumented migrants. They plan to return home before retirement—they’ve already built houses back in China.”* — Xiao (Female, 60+, Mainland China)

*“Fujianese people abroad care only about one thing: making money. Integration doesn’t even cross their mind.”* — Glasses (Male, 55+, Mainland China)

*“I need to save for my son’s wedding and house. In China, parents still carry these responsibilities, even when the child has grown.”* — Amy (Female, 55+, Mainland China)

*“My duty was to raise my child. Now that she’s graduating, I’ll return to China for retirement.”* — Zhang (Male, 55+, Mainland China)

As a result, many Chinese migrants remain functionally self-sufficient within “ethnic enclaves”. In urban areas like Chinatowns or predominantly Chinese suburbs, access to familiar services—such as Mandarin-speaking doctors, Chinese supermarkets, and bilingual legal support—reduces the need for language acquisition or broader social integration.

*“I don’t have many friends in the UK, and those I do have are mostly Chinese. Especially during the pandemic, since my last holiday in February 2020, I’ve barely left my small town. For elderly people, the virus was a major concern, so I avoided any external contact.”* — Xiao (Female, 60+, Mainland China)

*“I lived in Australia for years—there are even more Chinese there. You can do everything in Mandarin: groceries, banking, even getting legal help.”* — Li (Female, 55+, Mainland China)

This form of “ethnic self-sufficiency” reinforces traditional practices such as frugality, re-use, and self-reliance—behaviors that align with low-carbon living even if they are not environmentally motivated. By reducing exposure to consumerist norms and bypassing mainstream cultural expectations, older migrants maintain habits rooted in economic prudence and cultural continuity.

### 4.2. Awareness–Action Gap Across Generations:
Why Younger Migrants Understand Policies Better, but Older Migrants Practice More


This section addresses Research Question 2, which explores the apparent paradox between younger migrants’ higher awareness of environmental policies and older migrants’ stronger engagement in low-carbon practices. The analysis reveals how generational differences shape environmental knowledge, values, and everyday behaviors.

#### 4.2.1. Information Access and the Awareness–Practice Gap

Younger and older Chinese migrants demonstrate distinct ways of accessing environmental information and engaging with sustainable behavior. These differences shape their understanding and adoption of climate-related practices.

Younger Migrants (Second-Generation and International Students)

Tend to access environmental information more easily through digital media, formal education, and social networks.Display higher awareness of UK climate policies and sustainability discourse due to stronger English skills and schooling in the UK.However, there is often a gap between their environmental knowledge and everyday behavior.

*“My son often sends me news articles and links to keep me updated. That’s how I first learned about ‘low-carbon’ policies. But in China, this topic still isn’t discussed much.”* — Rui (Female, 75+, Mainland China)

*“The next generation definitely has better English. Many of them were born here and fully understand the local system.”* — Huang (Male, 70+, Hong Kong)

Older Migrants (First-Generation, Aged 55+)

Rely on Chinese-language media and word-of-mouth in ethnic communities for information.Engage in low-carbon behaviors due to deeply rooted frugality and resource-conscious habits, rather than policy awareness.The walking interviews revealed a range of such practices embedded in daily life.

Examples of Frugal, Low-Carbon Practices

1.Conscious Spending and Minimal Waste

*“I only buy clothes when I really need them, or during big discounts. I hate waste. If something can still be used, I use it or donate it.”* — He (Female, 60+, Malaysia)

*“If you go to the supermarket before it closes, bread that costs £1 can go down to 20p. Vegetables too. You just need to know where to look.”* — Jian (Female, 65+, Mainland China)

2.Energy Efficiency

*“We don’t turn on the heating all day. We keep curtains shut to preserve heat and only adjust the temperature when needed. We even use firewood when possible.”* — Zhang (Male, 55+, Mainland China)

3.Composting and Rainwater Use

*“We compost fruit peels and collect rainwater in barrels for our plants. I reuse laundry water to mop the floor. It’s just how we live.”* — Xiao (Female, 60+, Mainland China)

In the walking interviews, participants eagerly demonstrated their repurposing skills at home. As illustrated in Figure 2, the photo on the left shows a creative gardening setup by Xiao, an elderly female participant passionate about horticulture. She transformed a discarded bathtub and sink into makeshift planters, exemplifying how older migrants engage in low-carbon gardening practices using salvaged materials.

The right image captures a clever frugality practice by another participant, Yi. She collected small remnants of soap—typically too soft and slippery to use—wrapped them in worn-out stockings, and used the mesh to create foam while preserving every last bit of the soap. This practice reflects an embodied knowledge of resourcefulness rooted in cultural norms of thrift and waste avoidance.

4.Repurposing Clothes

*“If something doesn’t fit anymore, I alter it. If it’s damaged but still usable, I fix it and wear it again.”* — Rui (75+, Mainland China)

Sewing and mending were also frequently mentioned as frugal and culturally significant practices among older female participants. Many recalled learning to sew from their mothers or older sisters during their youth in China, often out of necessity. For example, younger siblings would typically wear hand-me-downs from older children, and homemade clothes were standard.

Despite living in a consumer-rich society today, participants expressed a sense of pride in maintaining these skills. As shown in Figure 3, Rui, a grandmother participant, is seen designing and stitching a prom dress for her granddaughter using leftover fabric and sewing patterns. For Rui, sewing was not only a means of saving money but also a way to transmit care and family values through handmade labor. These practices, while no longer essential in an economic sense, remain meaningful expressions of cultural continuity and sustainable living.

5.Avoiding Food Waste

*“I never waste food. I use what I have and fix what I can. Christmas sales? I don’t buy unless I really need something.”* — Huang (Female, 60+, Taiwan)

6.Reusable Bags and Household Recycling

Participants often repurposed materials as part of their daily sustainability practices, driven by both thrift and creativity. As illustrated in Figure 4, the image on the left shows a collection of used paper gift bags, neatly stored for reuse during shopping or gift-giving. On the right, a handmade shopping bag has been sewn from a pair of old jeans. This particular item was created by Ye, a male participant in his 60s from Mainland China, who proudly shared:


*“We haven’t bought plastic bags in years. I sew shopping bags from old clothes and even my children’s school uniforms.”*


This act of reusing not only avoids single-use plastic but also extends the lifespan of textile materials that might otherwise become waste. Such practices, though often overlooked in formal sustainability discourse, reveal a deep cultural ethic of reuse, repair, and responsibility that underpins many older migrants’ low-carbon lifestyles.

7.Growing Vegetables at Home

As shown in Figure 5, home-grown vegetables and fruits were an important aspect of many participants’ lives. The image displays the produce from Jian’s garden, which includes tomatoes, apples, and various herbs. Jian not only cultivated food for personal use but often shared surplus produce with neighbors and friends. During the interview, she expressed ongoing enthusiasm for experimenting with new crops, saying:

*“I grow chives, coriander, and spinach—more than we can eat. I share with friends and enjoy seeing them happy.” “I think gardening is something very Chinese. In rural China, every household had chilies, eggplants, and vegetables growing in their yard. It’s not about money—of course, we can afford supermarket food—it’s about making use of our time meaningfully and staying healthy in retirement.”* — Jian (Female, 65+, Mainland China)

Gardening served as more than a cost-saving measure; it was a way of sustaining cultural traditions, fostering neighborly ties, and maintaining a sense of purpose and physical activity.

*“Homegrown food just tastes better. Since retiring, I hardly buy vegetables anymore.”* — Ye (Male, 60+, Mainland China)

As illustrated in Figure 6, participants not only grew their own vegetables but also actively incorporated them into home cooking. The left image displays a harvest of squashes and gourds, while the right image shows handmade pumpkin pancakes prepared by Ye, a male participant in his 60s from mainland China. During the interview, Ye also proudly shared:


*“Vegetables grown by yourself just taste different. The process—from planting to cooking—makes you appreciate food more.”*


#### 4.2.2. Historical Memory and the Habit of Saving

The generational divide among Chinese migrants is shaped by starkly different historical experiences. Older generations grew up during times of extreme scarcity, rationing, and hardship—circumstances that instilled enduring habits of thrift, reuse, and anti-waste sensibilities. These behavioral patterns persist across time and space, shaping their everyday decisions even after migration.

Older adults internalized frugality through lived experience—often linked to famine, war, and national austerity campaigns.Younger generations, raised during periods of economic reform and expansion, had more access to material goods and developed different consumption habits, even though they may still respect the values of thrift.

*“As my parents say, ‘Each generation is less frugal than the last.’ My children don’t waste on purpose, but they haven’t lived through hardship, so they don’t see the value in small savings.”* — Leng (Female, 60+, Mainland China)

*“My grandmother lived through starvation. She would never waste a single grain of rice. That kind of experience stays with you.”* — Huang (Male, 70+, Hong Kong)

*“Now we have Chinese supermarkets everywhere. Back in my day, my mother had to send me parcels from China every month.”* — He (Female, 60+, Malaysia)

*“The older people in my village grew up starving. In Luoyang, people didn’t say ‘Have you eaten?’—they asked ‘Have you had soup?’ That’s how poor we were. Even now, the elderly always finish their meals and save everything. Their children don’t understand why they keep so much ‘useless junk.’”* — Xiao (Female, 60+, Mainland China)

*“Older people never waste food. We younger ones never experienced that kind of hunger, so we don’t really understand. But hearing their stories, it makes sense—they lived through starvation in the 1950s. Those were very hard times.”* — Jian (Female, 65+, Mainland China)

These accounts highlight how memory of hardship functions as a “moral anchor and behavioral driver”, reinforcing low-consumption lifestyles that align with sustainability. Even when living in materially more secure environments, older migrants continue to embody a frugal ethic passed down through family and community. This intergenerational gap in environmental behavior is less about ideology, and more about “lived memory and value transmission”.

#### 4.2.3. Spending Habits and Consumer Culture

Patterns of financial behavior among Chinese migrants reflect a clear generational divide shaped by socialization, migration timing, and cultural exposure. Younger migrants—especially those who arrived during or after China’s reform era—are more exposed to Western consumerism, often embracing credit use, lifestyle consumption, and instant gratification. Older migrants, by contrast, tend to maintain more conservative financial habits rooted in thrift and delayed gratification.

*“Today’s young people spend too freely—luxury handbags, fast fashion. Our generation saves first, spends later.”* — Huang (Female, 60+, Taiwan)

*“Western thinking is catching on—borrow now, enjoy today. Mortgages make sense sometimes, but saving gives peace of mind.”* — Ye (Male, 60+, Mainland China)

These contrasting approaches to spending also inform environmental behavior. While younger migrants may be more environmentally informed, their higher material consumption can contradict low-carbon ideals. Older migrants, on the other hand, practice “quiet sustainability” through habits of moderation and resource conservation. As such, “economic values—whether shaped by cultural upbringing or external consumer pressures—play a critical role in structuring sustainable or unsustainable practices across generations.”

### 4.3. Frugality as Cultural Practice

This section explores the historical, cultural, and economic underpinnings of frugality as a sustainable lifestyle among older Chinese migrants. Their low-carbon behaviors are not necessarily framed in environmental terms, but emerge from lived experience, moral values, and structural constraints. Four key subthemes illustrate how thrift is practiced and maintained across time and contexts.

#### 4.3.1. Confucian Values and Social Norms

Beyond survival, thrift is also a cultural virtue. Rooted in Confucian teachings, frugality is seen as morally appropriate and socially responsible. These values are passed down through family and reinforced by community expectations.

*“Traditional Chinese culture values loyalty, collectivism, and personal sacrifice. Frugality is part of this—it’s about being responsible to your family and community.”* — Zhang (Male, 55+, Mainland China)

*“We were taught not to waste anything—don’t use paper unnecessarily, don’t buy things you don’t need. It’s just a habit from our generation.”* — Xiao (Female, 60+, Mainland China)

#### 4.3.2. Economic Conditions and Welfare Adaptation: Between Scarcity Memory and Present Stability

Although many older Chinese migrants now enjoy greater financial stability in the UK, their frugal behaviors are shaped by a dual awareness: the memory of past hardship and a comparative understanding of current economic conditions in China. Rising costs of housing, healthcare, and education in their home country continue to validate thrift as a rational long-term strategy—especially for those supporting family members across borders.

*“Back in China, I was laid off. But here in the UK, I can rely on my husband’s pension. I don’t talk to my old friends in China anymore—their lives are too different. I can travel and enjoy life, while they’re still struggling just to get by.”* — Piao (Female, 60+, Mainland China)

*“If I were still in China, I’d have only 2,000–3,000 yuan a month in pension. That’s not enough to live on. But here, the UK government supports us. I can retire without fear.”* — Xiao (Female, 60+, Mainland China)

Upon migrating, many participants adapted to the UK’s more generous welfare system. Pensions, healthcare access, and disability support reduced their immediate financial anxieties and created a sense of long-term security.

*“When my husband got brain cancer, the UK system really helped us. They immediately gave us disability status, plus £500 per month, daily nurse visits, and transport subsidies. It’s not just words—it’s a real safety net.”* — Xiao (Female, 60+, Mainland China)

Despite improved material conditions, participants largely maintained frugal habits. Saving, reusing, and avoiding waste were not simply economic choices, but "culturally ingrained responses to insecurity".

### 4.4. Continuity and Change After Migration

This section explores the dynamic interplay between change and continuity in the lives of older Chinese migrants. While many aspects of daily life—such as language, diet, and consumption—have adapted to the host environment, deeply rooted cultural values, particularly frugality and identity, remain remarkably resilient.

#### 4.4.1. External Adaptation: Language, Diet, and Consumption

Upon migration, older Chinese migrants naturally adjust their lifestyles in response to changes in physical environment, access to familiar resources, economic conditions, and social networks.

##### Environmental Adaptation

Limited access to traditional ingredients has led many to modify their eating habits, often blending Chinese and Western food practices—especially within mixed-nationality households.

*“There were no Chinese supermarkets back then. Every weekend, I had to travel from Oxford to London’s Chinatown with a full backpack just to bring home enough ingredients. Now, Chinese food is trendy—even British supermarkets stock condiments. But back in those days, we couldn’t even find spring onions or ginger.”* — Jiao (Female, 60+, Mainland China)

*“I eat more Western food now. But when I crave Taiwanese or Chinese dishes, I cook them for myself. My husband sticks to his usual Western meals—we each enjoy our own preferences.”* — Huang (Female, 60+, Taiwan)

##### Social Influence

Cross-cultural contact has also reshaped cooking and language practices. Many participants noted a growing openness toward Western norms and convenience-focused lifestyles.

*“At Chinese gatherings, people used to bring homemade food. But now, especially in mixed-nationality families, many bring store-bought items from British supermarkets. At first, I thought they’d taste bad—but they’re actually quite good. We used to make everything from scratch, but now I find supermarket options more convenient.”* — Fa (Female, 60+, Mainland China)

*“Working in a multicultural environment really broadens your perspective. You learn how different cultures communicate and how to collaborate beyond just your own circle.”* — He (Female, 60+, Malaysia)

*“To improve my English, I joined a local club where I chat, cook, and have tea with British people. It helped me speak more naturally and get used to their communication style.”* — Amy (Female, 55+, Mainland China)

##### Economic Shifts and New Spending Patterns

Many migrants experience increased income or financial stability after moving abroad. This enables new consumption habits, often contrasting sharply with their past lifestyles in China.

*“In the UK, I’ve bought a lot of luxury items—but always on sale. Back in China, I wouldn’t even consider these things. But now my kids are grown up, and I have the freedom to enjoy life. People say it’s materialistic—I call it pleasure.”* — Li (Female, 55+, Mainland China)

Figure 7 Luxury items shared by participant Li, who purchased discounted branded clothing and accessories from Bicester Village.

*“During the pandemic, I couldn’t travel, so I had extra money. I even treated myself to beauty services for the first time. I shop with my neighbours now—sometimes I buy too much and just give it away if I don’t like it later.”* — Piao (Female, 60+, Mainland China)

*“Here, if your savings go over £15,000, your benefits are reduced. So people say, ‘spend aggressively’ to keep entitlements. It’s the opposite of China, where you save in case of illness. With free healthcare here, you don’t need to stockpile savings.”* — Piao (Female, 60+, Mainland China)

These behaviors reflect more than just increased purchasing power—they also signal a psychological shift. For some, spending becomes a way to reclaim pleasure and autonomy after decades of sacrifice and financial caution. In this context, consumption may function as a “reactive or compensatory response” to past deprivation and the abrupt contrast between two welfare systems. While these practices may appear to contradict earlier frugality, they also highlight the complexity of migrant adaptation.

#### 4.4.2. Cultural Continuity: Deeply Rooted Values Persist

Despite visible lifestyle changes, core cultural values—particularly frugality and traditional identity—continue to shape behavior, underscoring the resilience of early life experiences and family teachings.

*“Frugality is part of my culture, and I’ve always kept that belief. I may wear jeans now, but during Lunar New Year, I still put on traditional Chinese clothes. I’ve adapted to Western life in some ways, but Chinese values remain.”* — He (Female, 60+, Malaysia)

*“We Chinese save for emergencies. Many British people rely on credit—they spend first and worry later. That kind of lifestyle creates stress, especially now with inflation and strikes.”* — Rui (Female, 75+, Mainland China)

*“Even after so many years in the UK, I haven’t changed my financial habits. I still save. Spending next month’s salary in advance just feels wrong to me. I guess it’s an Asian thing—we need that sense of security.”* — Huang (Female, 60+, Taiwan)

Despite visible lifestyle changes following migration, many participants continue to uphold core cultural values, especially frugality, family responsibility, and ethnic identity. These values—shaped by early life experiences, Confucian moral teachings, and intergenerational socialization—remain stable even in the face of new economic opportunities and host society norms.

## 5. Discussion

This study aimed to understand how older Chinese migrants in the UK perceive and respond to cultural integration, intergenerational environmental differences, and the significance of frugality in daily sustainable living. By employing walking interviews, we collected personal narratives that highlight sustainability not merely as a conscious decision or policy-informed action, but often as deeply ingrained cultural habits and lifestyles shaped by historical experiences and cultural norms.

Despite increasing emphasis on achieving net-zero transitions, sustainability discourses and climate policies frequently focus on majority populations and technology-driven interventions. Consequently, older migrant groups—particularly those with limited language proficiency or digital access—are often marginalized in sustainability agendas.

Our findings show that many older migrants continue “doing what they have always done”—repairing, saving, and reusing—not as acts of environmentalism, but as habitual practices rooted in cultural norms. Mainstream sustainability narratives, which often prioritize green technology and consumer choice, typically overlook these subtle yet significant practices. Thus, our research contributes to calls for a broader, more inclusive understanding of sustainable behaviors. Although models such as the Theory of Planned Behavior (TPB) [32] and Value-Belief-Norm (VBN) theory [19] offer valuable insights, they often neglect the influences of habitual actions, community expectations, and cultural memory. Practice-based theories, emphasizing how actions are learned and repeated over time, provide a more grounded perspective on migrant sustainability [21].

It is also notable that many sustainability policies presume certain educational levels, language proficiency, or digital literacy. This assumption unintentionally excludes populations already practicing sustainability outside mainstream frameworks. Older Chinese migrants, as our study indicates, rarely engage in formal environmental initiatives yet still lead sustainable lives characterized by frugality, home gardening, and material reuse based on long-held cultural values.

### 5.1. Key Findings and Interpretation

#### 5.1.1. Frugality: Cultural Legacy or Economic Response?

The participants often described their frugal practices as cultural expectations and personal satisfaction. Although historical economic hardships clearly shaped these behaviors, many continue to live frugally even after achieving greater financial security in the UK. For some, this behavior stems from persistent feelings of vulnerability, while others consider it simply the “correct” way of living. These insights remind us that frugality is not always passive—it can also be proactive and aspirational.

Interestingly, we observed moments of contrast where participants occasionally indulged in luxury goods, particularly when they felt more financially secure. These changes illustrate a layered logic: old habits persist, yet new behaviors emerge without fully replacing the old. They represented a temporary adjustment or psychological response to improved living conditions.

#### 5.1.2. Between Tradition and Comfort: Negotiating Consumption

The coexistence of frugality and indulgence within the same households may appear contradictory but typically reflects practical compromises. Many older migrants retained their traditional values while adapting to new norms regarding comfort and convenience. This balancing act demonstrates that cultural identity is not static; it evolves through negotiation with place, available resources, and social expectations. Migrants often adapt their behaviors while maintaining core cultural beliefs.

#### 5.1.3. Social Exclusion and Quiet Sustainability

Several participants shared experiences of racism or social exclusion, particularly during their early years in the UK. In response, many turned inward, increasingly relying on their ethnic communities. These ethnic networks—through Chinese supermarkets, social clubs, and WeChat groups—became essential spaces for mutual support and knowledge exchange. Within these communities, reuse, repair, and conservation practices became common topics of discussion. Although not explicitly identified as environmentalism, these practices closely align with sustainable living, emerging clearly through an in-depth examination of their daily lives.

#### 5.1.4. Awareness Without Action? Generational Sustainability Tensions

The participants noted generational differences based on their observations. Many believed that younger Chinese migrants had greater education and environmental awareness, but were inconsistent in sustainable practices. In contrast, older migrants rarely used “green” terminology, but consistently practiced low-carbon, frugal lifestyles. The perceived awareness–action gap observed in this study presents a challenge to the Value–Belief–Norm (VBN) theory, which assumes a linear progression from environmental values and beliefs to pro-environmental behaviors. Our findings indicate that high levels of awareness of policy do not necessarily translate into consistent sustainable action. Some migrants, for instance, demonstrated high-consumption behaviors, suggesting the influence of convenience or psychological adjustment following migration. Conversely, several older participants lacked formal environmental terminology but continued to engage in low-carbon practices driven by long-standing values and habitual norms. These insights highlight the value of complementing the VBN framework with practice theory to better capture the complexity and embodied nature of sustainability behaviors in migrant contexts. Our findings challenge this assumption, highlighting the importance of studying sustainability as lived experience rather than merely formal education.

#### 5.1.5. Rethinking Inclusion in Environmental Narratives

Mainstream definitions of environmental behavior frequently ignore quiet, habitual forms of sustainability deeply rooted within working class, migrant, or elderly communities. Our study advocates reframing these narratives to acknowledge that individuals can practice significant sustainability without explicitly labeling themselves “green”. Policies and programs should reflect this reality by valuing reuse and frugality, engaging trusted local networks, and communicating in terms resonating with daily life, thereby enhancing inclusion of minority and marginalized groups.

#### 5.1.6. Interplay of Cultural, Economic, and Social Factors

While this study primarily emphasized cultural influences on sustainable practices, our findings also highlight the role of economic and social factors. For example, concerns about rising utility costs were frequently mentioned as drivers for energy-saving behaviors, suggesting that frugality is shaped both by cultural memory and financial necessity. Additionally, participants’ limited access to broader social networks—due to language barriers or perceived exclusion—further encouraged resource sharing and reuse within tight-knit ethnic communities. These observations underscore the need to understand sustainable behaviors as outcomes of multiple overlapping influences, including cultural identity, socio-economic vulnerability, and structural constraints.

### 5.2. Reflections on the Walking Interview Method

Walking interviews enabled participants to share their experiences naturally and at their own pace. For many, navigating familiar neighborhoods facilitated openness. The participants often referenced tangible examples—homegrown plants, visited shops, reused items—that concretized abstract sustainability concepts.

However, the method presented challenges, such as mobility constraints, weather conditions, and public noise disturbances. Despite these challenges, walking interviews provided opportunities to obtain richer, more authentic insights into lived sustainability practices.

### 5.3. Limitations and Future Research

This research exclusively focused on older Chinese migrants in urban Greater London. Although thematic saturation was reached, our findings may not be generalizable to rural areas or different migrant groups. Additionally, insights into younger generations were derived indirectly through older participants’ perspectives rather than direct interviews.

Future studies should include diverse age groups and geographic regions to better understand intergenerational transmission and evolution of sustainability practices. Additionally, exploring influences from gender roles, religious institutions, or digital media within migrant families could yield deeper insights.

### 5.4. Implications for Policy and Practice

Our findings indicate several missed opportunities in current environmental policy frameworks, particularly regarding engagement with older migrant populations. These groups often exhibit sustainable behaviors rooted in cultural values and past experiences, but remain excluded from mainstream programs due to linguistic, cultural, or digital barriers.

To address this gap, we propose the following actionable recommendations:Recognize and integrate migrant frugality into sustainability planning. Instead of viewing older migrants’ thrift-based behaviors as outdated, policy frameworks should consider them as assets. Practices such as repair, reuse, and resource-sharing can be promoted through culturally sensitive campaigns.Partner with ethnic institutions for outreach. Trusted community nodes—such as Chinese community centers, churches, Chinese-language newspapers, and grocery stores—are critical entry points. Environmental messages can be effectively distributed through local Chinese organizations, WeChat groups, Xiaohongshu (RED), and even TikTok videos in Mandarin/Cantonese.Leverage intergenerational collaboration. Programs that encourage younger migrants to act as policy interpreters or digital facilitators for older family members can bridge the awareness–practice divide. These initiatives can build mutual respect and promote sustainability within households.Embed climate communication within culturally resonant narratives. Messages framed around frugality, family responsibility, and care for future generations may resonate more than abstract concepts like “carbon footprint” or “net-zero targets.” Policymakers should adapt language and framing to local cultural contexts.Identify and support community sustainability champions. Migrants already practicing “quiet sustainability” could serve as peer educators. Informal influencers within ethnic networks can lead workshops or storytelling events to share lived experiences of low-carbon lifestyles.

Furthermore, observed behaviors—such as growing vegetables at home, creative reuse of materials, and avoiding food waste—indicate that cost-effective interventions (e.g., energy-saving grants, insulation workshops, DIY repair guides) may be more successful when delivered through culturally appropriate channels.

Finally, integrating these practices into formal sustainability programs requires a shift away from a one-size-fits-all model. Planners and practitioners should adopt participatory approaches that value localized, culturally embedded knowledge, ensuring that sustainability transitions are truly inclusive and equitable.

### 5.5. Conclusions

Older Chinese migrants are often excluded from mainstream narratives of sustainability and climate action. Yet, as this study demonstrates, they embody forms of everyday low-carbon living that are rooted in culture, shaped by hardship, and preserved across borders. Their frugality, conservation, and community-centric practices offer a rich, often untapped foundation for inclusive environmental policy.

By respecting their cultural identities, acknowledging their contributions, and addressing barriers to participation, policymakers and practitioners can ensure that sustainability efforts do not overlook these quiet stewards of resourcefulness. As the climate crisis demands collective action, the wisdom of this generation offers both a moral compass and practical guide for living lightly on the Earth.

## Figures and Tables

**Figure 1 ijerph-22-00832-f001:**
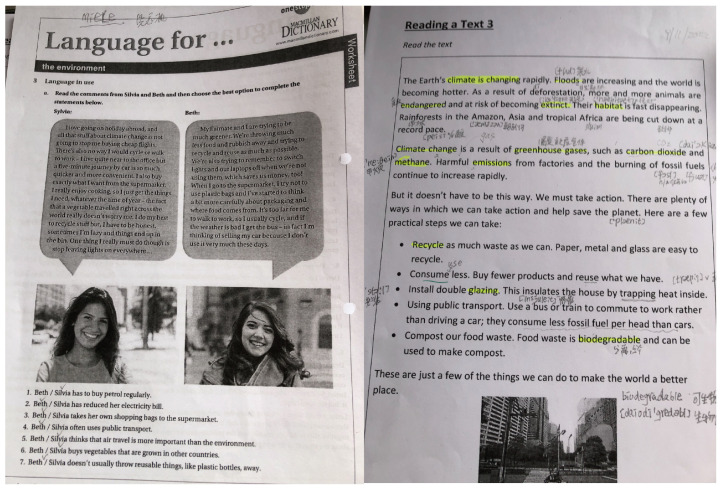
English learning materials used by Rui during language self-study sessions. Some handwritten notes in Chinese are visible, reflecting the participant’s bilingual approach to language learning.

**Figure 2 ijerph-22-00832-f002:**
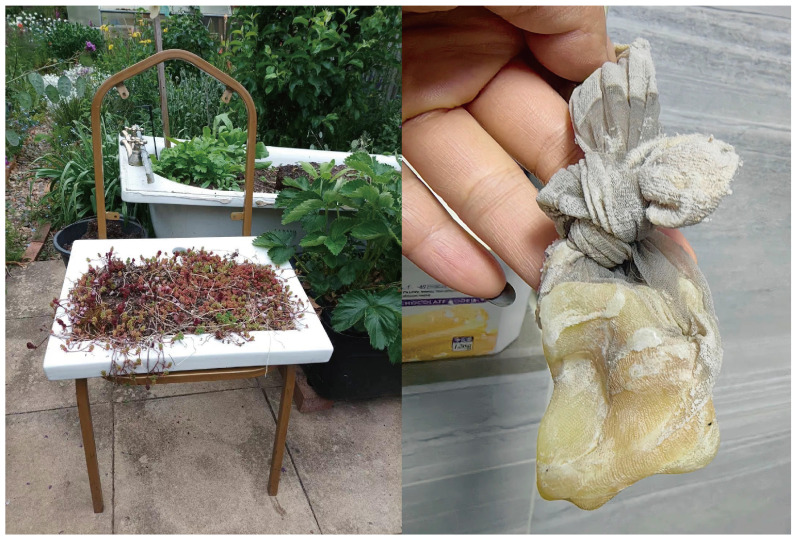
Waste repurposing practices: (**left**)—discarded bathtub used for planting; (**right**)—soap remnants wrapped in stockings.

**Figure 3 ijerph-22-00832-f003:**
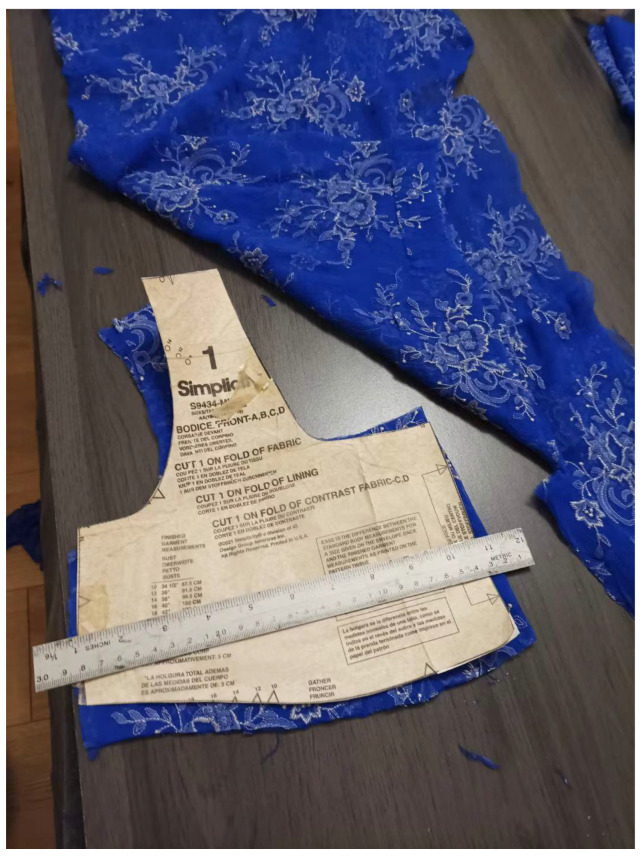
Modified clothing made by participant.

**Figure 4 ijerph-22-00832-f004:**
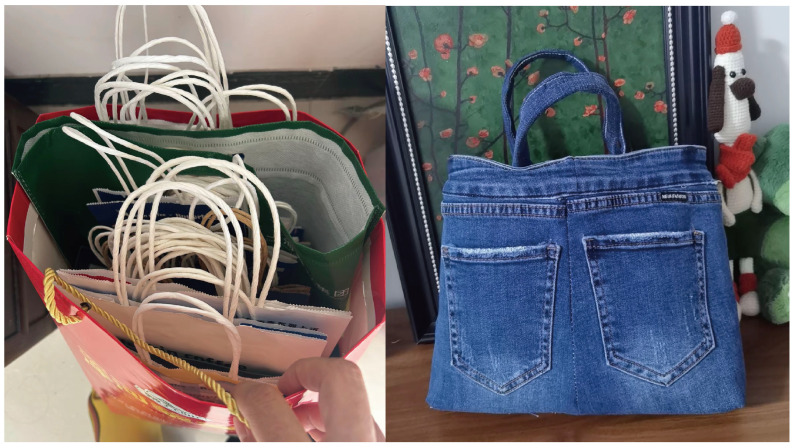
Repurposed shopping bags made from old jeans and uniforms.

**Figure 5 ijerph-22-00832-f005:**
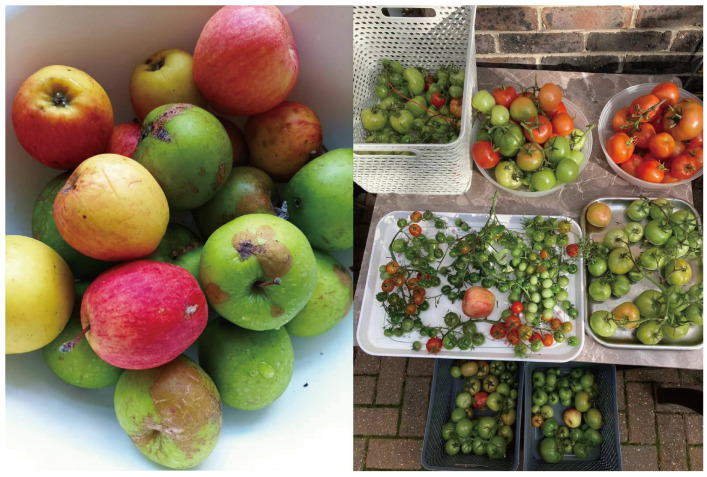
Vegetables and herbs grown by participants.

**Figure 6 ijerph-22-00832-f006:**
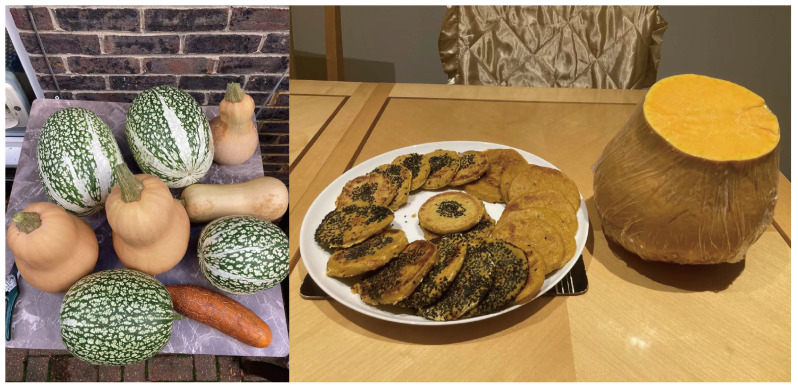
Cooking with vegetables participants grow.

**Figure 7 ijerph-22-00832-f007:**
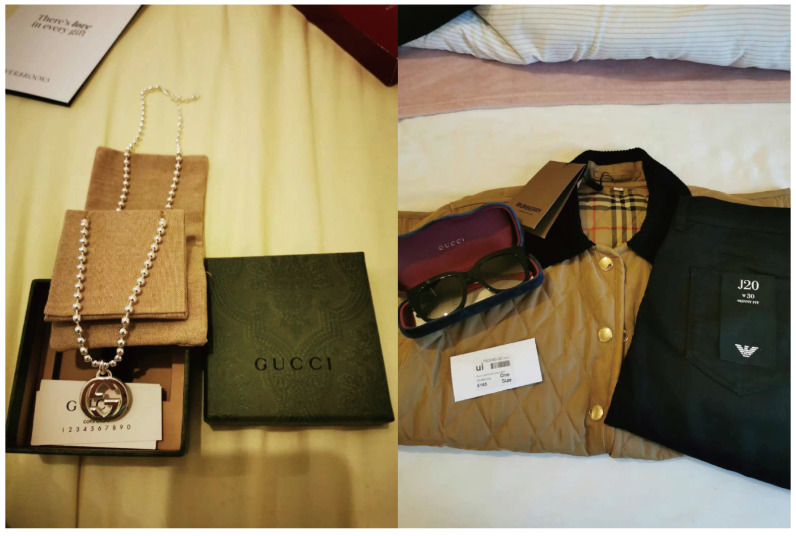
Luxury goods and shopping bags shared by the participants.

**Table 1 ijerph-22-00832-t001:** Summary of participant characteristics (N = 20).

Category	Distribution
Gender	13 Female, 7 Male
Age Range	55–64 (10), 65–74 (6), 75+ (4)
Migration Background	- Former professionals/international students (4) - Marriage migrants (3) - Labor migrants (7) - Family dependents (6)
Low-Carbon Behavior Type	- Environmental Pioneers (3) - Frugal Practitioners (9) - Passive Adapters (5) - Isolated Participants (3)
Ethnic Origin	- Mainland China (16) - Hong Kong (1) - Malaysia (2) - Taiwan (1)
Cross-national Family Background	4 participants

## Data Availability

The data relating to this study can be obtained from the first author.

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
