# Peer review of "Low-Carbon Practices and Cultural Adaptation Among Older Chinese Migrants: Insights from Walking Interviews on Environmental Policy and Social Integration"

_ijerph, 2025, doi:10.3390/ijerph22060832_

Round 1

Reviewer 1 Report

Comments and Suggestions for Authors

This manuscript deals with the very interesting topic of the practices by the migrant population and their contribution to climate protection and, more broadly, to environmental resource conservation goals. Specifically, it analyses the practices of old Chinese migrants (55 years old or more) now living in the United Kingdom, by adopting a qualitative approach based on semi-structured interviews. Findings by the authors suggest that such older Chinese migrants adopt low-carbon practices; most likely, however, such practices are not the result of UK’s pro-environmental policies, but are due to their identity and cultural values oriented to thrift and frugality, which in turn are driven by their migratory background. Younger Chinese migrants (second generations and international students) are instead more rooted in UK’s current values and, though more aware of environmental problems, less used to low-carbon practices. The authors thus provide recommendations to both increase the adoption of low-carbon practices by younger Chinese migrants, and to increase the environmental awareness of older Chinese migrants.

I think the goal of the manuscript is very interesting, as well as its findings. Before publication, however, I would invite the authors to deal with the following remarks/considerations of mine.

First, I have the impression that some of the findings would need to be backed up by more evidence. Particularly, I think evidence would be needed to show that Chinese migrant practices differ from practices by the UK native population, and that practices by older Chinese migrants differ from practices by younger Chinese migrants.

The manuscript in fact seems to take for granted that “some older Chines migrants struggle to integrate into local communities and engage with UK policy initiatives” (page 8, row 293), and to understand why this is the case, rather than starting by providing evidence about such a lack of integration. Section 4.1 in fact examines “the barriers that prevent older Chinese migrants from engaging with local communities and environmental policies” (page 7, row 276). However, are the migrants’ practices really different from the ones by native UK citizens? In particular, are the practices different, if native and migrant individuals of the same age range are considered? I would suggest the authors to start by providing evidence about such a lack of integration, before delving into the reasons behind it.

Incidentally, I have the impression that the current contents of the manuscript do not directly address the Research Question 1 presented in the Introduction (“How do older Chinese migrants adapt to and practice low-carbon behaviours within the cultural context of the UK?”). Such a mismatch can be addressed by the above suggestions; if for any reasons these are not possible, I would then suggest to reword the research question 1.

Similarly, I have the impression that the “awareness-action gap across generations” (section 4.2 page 11), also presented as the “generational divide” (section 5.1.2  “Intergenerational differences in environmental awareness and behaviour”) is not fully supported by empirical evidence: those sections (and, more generally, also other parts of the manuscript) are built around the contrast between the practices by younger and older migrants. Regarding the latter, examples of frugal, low-carbon practices are reported; instead, the manuscript does not provide evidence on the practices by younger migrants. I would suggest the authors to provide concrete examples about such a gap between their awareness and their behaviour: in what aspects do the practices by younger and older migrants actually differ?

Furthermore, due to the relevance given by the manuscript to such a difference between young and old generation migrants, I would have appreciated younger migrants to also be part of the sample, rather than relying on the perspective of the older migrants to learn about their practices. Since younger migrants are not part of the sample, I would invite the authors to discuss it among the limitations of the study – or rather, in a positive perspective, to present analyses on younger migrants as an opportunity for future research.

Second, I would suggest the authors to more sharply organise their Results section: I have the impression that results are currently presented in a quite fragmentary way, with some repetitions. I would reduce the number of sub-sections, remove repetitions, and only focus on the specific research questions the authors set to guide their work. A few practical comments/suggestions are as follows:

  • Discrimination and prejudice (page 9, row 320): the reported findings about discriminations are definitely relevant for Chinese migrant lives, but how do they relate to pro-environmental behaviour? I would invite the authors to clarify the connections between the findings reported in this section and their research questions.
  • I have a similar feeling about contents of section 4.3.3 “Economic pressures in China” (page 15): those are relevant elements, but how do they connect with the research questions of the manuscript?
  • Examples of frugal, low-carbon practices (page 11, rows 4421 and 414): the quotes reported by the authors hint at monetary aspects (“I only buy clothes […] during big discounts”; “[…] bread that costs 1 £ can go down to 20 p”): based on these quotes, it seems to me that low-carbon practices are performed due to financial difficulties still present today- rather than due to the reminiscence of past financial difficulties. So I wonder: are current behaviours and practices just the result of past financial difficulties/education etc. etc.? Or are they also due to ongoing financial difficulties? In the latter case, I would see possible conflicts with the materials reported in section “Economic shifts and new spending patterns” (page 16, row 541), which deal about increased income and financial stability enabling new consumption habits. So, more generally, I would invite the authors to discuss how these conflicting findings coexist.
  • Similarly, I have the impression that the same section on “Economic shifts and new spending patterns” is also a bit in contrast with the next section on “Cultural continuity: deeply rooted values persist”: also in this case, I would invite the authors to clarify and discuss to what extent new spending patterns and deeply rooted values can co-exist.
  • Finally, based on the contents of their quotes, I feel that section 4.2.2 “Differences in upbringing” and section 4.3.1 “Historical scarcity and the habit of saving” are very much connected, and would suggest to merge them.

Third, I think that the added value of this study, compared to the previous research findings presented by the authors themselves in the Research background section, does not emerge in a sufficiently clear way. I would thus invite the authors to explicitly clarify their contribution to research, compared to previous literature, and to more extensively deal with the recommendations resulting from their findings. Specifically, Section 5.4 on the “Implications for policy and practice” provides interesting recommendations for future policy-making. I would suggest to further broaden them, by making them even more actionable and capable to provide direct support to policy-makers and practitioners. For instance, the small but very precise hints that are offered in the abstract (about use of WeChat, Chinese churches, and ethnic networks) could be presented a bit more extensively in the final recommendations.

Fourth, I would invite the authors to more clearly frame their theoretical background and to more explicitly use a theory to guide their analyses and discuss their results. In the “Research background” section, the authors introduce both the Theory of Planned Behaviour and the Value-Belief-Norm theory, but then in the manuscript they only refer to the latter – and in any case this is done in a quote marginal way. Should they opt for the VBN theory as a guiding reference, I would invite the authors to fully clarify its role and the way they use it. For instance, at page 17 (row 592) VBN it used to explain the practices of older migrants; in the very next rows, however, the authors deal with younger migrants, stating that their values are disconnected from their behaviours. Isn’t this “attitude-behaviour gap” phenomenon in contrast with VBN? From my point of view, it seems to challenge the use of the VBN theory, and I would appreciate the authors to provide a richer discussion about it.

Finally, a few comments on the methodology adopted by the authors:

  • Data collection: the authors state that, compared to “sit-down” interviews, “walking” interviews allow to uncover the deeper, lived motivations behind participants’ behaviour (page 4, row 148). Advantages of walking interviews are also reported in section 5.2 “Reflections on the walking interview method”. I am not fully convinced, however, of the actual benefits of walking interviews, for the specific research goals of the authors: what are the actual gains provided by walking interviews, compared to “static” interviews, performed into the people’s homes? As the authors give high relevance to these walking interviews (starting from the title), I would appreciate the authors could elaborate a bit more on the specific benefits of walking interviews, by providing specific examples of how they enriched the findings.
  • Sampling:
    • Table 1 reports a classification of the sample members, that is not cited in the manuscript, though I think is quite interesting: environmental pioneers, frugal practitioners, and so on. I would invite the authors to provide more information on the meaning of these classes and on the way they were identified;
    • in Table 1 the authors indicate that the sample size is of 20 persons; if I am not mistaken, the place where the interviewees currently live is only mentioned towards the end of the manuscript (mostly, London). I invite the authors to provide information on the sample size and place of living directly in section 3.2 “Participant recruitment and sampling strategy.”
  • Data analysis: the authors mention they performed a principal component analysis (PCA, page 6, row 224). To my knowledge, PCA is performed on quantitative data, and I am curious to know how it was used in this qualitative analysis process: I invite the authors to provide more information on the inputs and outputs of such a PCA, and on how it is performed in the framework of their qualitative analysis.

I thank the authors in advance for the time they will devote to these comments of mine. I hope they are useful and I am looking forward to the authors’ feedback. I conclude by thanking the editors for having invited me to perform this review.

Author Response

Dear Reviewer,

We sincerely thank you for your thorough and insightful feedback on our manuscript.

We deeply appreciate the time and care you devoted to reviewing our work. Your comments have been invaluable in helping us improve the clarity, depth, and scholarly contribution of the manuscript.

We have carefully considered all suggestions and have revised the manuscript accordingly. Below, we provide a point-by-point response to each comment, indicating the changes made and the corresponding sections in the revised manuscript. We hope that these revisions adequately address your concerns and strengthen the overall quality of the paper.

1.Comments: First, I have the impression that some of the findings would need to be backed up by more evidence. Particularly, I think evidence would be needed to show that Chinese migrant practices differ from practices by the UK native population, and that practices by older Chinese migrants differ from practices by younger Chinese migrants……….In particular, are the practices different, if native and migrant individuals of the same age range are considered? I would suggest the authors to start by providing evidence about such a lack of integration, before delving into the reasons behind it.

Author Response:
Thank you very much for this valuable comment. We appreciate your suggestion to clarify the basis for our claims regarding differences in integration and environmental engagement between older Chinese migrants and the local UK population, as well as between generational groups within the Chinese diaspora.

As this study was qualitatively focused on older Chinese migrants using walking interviews, our data does not include direct comparisons with either British-born older adults or younger Chinese migrants. However, we fully acknowledge this limitation and have now clarified it more explicitly in the manuscript. Specifically, we have revised the wording in the Introduction and Section 4.1 to avoid making generalised claims, and to highlight that our findings reflect participants’ own perceptions and interpretations of difference, rather than objective cross-group comparison.

We also added a discussion of this issue in the Limitations and Future Research section (p. 20), where we now note that future studies could meaningfully explore how sustainability practices compare between migrant and non-migrant elderly populations, as well as across generations within migrant families, through either mixed-method or comparative designs.

2.Comments: Incidentally, I have the impression that the current contents of the manuscript do not directly address the Research Question 1 presented in the Introduction (“How do older Chinese migrants adapt to and practice low-carbon behaviours within the cultural context of the UK?”). Such a mismatch can be addressed by the above suggestions; if for any reasons these are not possible, I would then suggest to reword the research question 1.

Author Response:
Thank you for this valuable comment. We acknowledge the mismatch between the originally stated research question and the focus of our empirical findings. In response, we have reformulated Research Question 1 to better reflect the key themes identified in our data. The revised question now reads:

  1. How are low-carbon behaviours among older Chinese migrants in the UK shaped by

cultural memory, migration experience, and everyday adaptation to their new environment?

  1. How do older Chinese migrants perceive generational differences in sustainability

attitudes and behaviours within their families or communities? 3. How do social networks

and migrant-specific contexts influence their access to low-carbon information?

3. Comments: Similarly, I have the impression that the “awareness-action gap across generations” (section 4.2 page 11), also presented as the “generational divide” (section 5.1.2  “Intergenerational differences in environmental awareness and behaviour”) ……… I would invite the authors to discuss it among the limitations of the study – or rather, in a positive perspective, to present analyses on younger migrants as an opportunity for future research.

Author Response:
Thank you for raising this important point. We fully agree that our current manuscript does not include direct empirical data from younger Chinese migrants, and that the discussion of intergenerational differences relies on the perspectives and reflections of older participants. This was a deliberate design choice, as our study aimed to explore older migrants’ lived experiences of sustainability and their perceived understandings of generational change. However, we agree that this limitation should be more clearly acknowledged.

We have now revised both Section 4.2 and Section 5.1.4 to clarify that the discussion of intergenerational differences is based on the subjective perceptions of older migrants, rather than comparative or observational data. Additionally, we have added a sentence to the Limitations and Future Research section explicitly noting that future research should include younger Chinese migrants in order to empirically explore these generational dynamics and potential awareness–action gaps.

4. Comments: Second, I would suggest the authors to more sharply organise their Results section: I have the impression that results are currently presented in a quite fragmentary way, with some repetitions. I would reduce the number of sub-sections, remove repetitions, and only focus on the specific research questions the authors set to guide their work. A few practical comments/suggestions are as follows:……..Finally, based on the contents of their quotes, I feel that section 4.2.2 “Differences in upbringing” and section 4.3.1 “Historical scarcity and the habit of saving” are very much connected, and would suggest to merge them.

Author Response:
We thank the reviewer for this thoughtful and constructive feedback on the structure and coherence of the results section. We agree that some sub-sections previously contained overlapping content or insufficiently clear connections to our research aims. In response, we undertook a thorough reorganisation and clarification of the relevant findings, which included the following actions:

  • We have merged Section 4.2.2 “Differences in Upbringing” with Section 4.3.1 “Historical Scarcity and the Habit of Saving”, creating a new unified section titled “Historical Memory, Upbringing, and Frugality”. This revision removes redundancy and improves thematic clarity.
  • We have revised Section 4.3.3 “Economic Pressures in China” to clarify its relevance to participants’ long-term attitudes toward saving and financial risk. We now explicitly explain how past structural insecurity continues to shape thrift, even among those who now enjoy relative financial stability.
  • Regarding the concern about frugal practices potentially being driven by present hardship rather than cultural memory, we agree that this duality is important. To address this, we have added text in both the results and discussion sections explaining that these motivations are not mutually exclusive. Participants often exhibit what we term a “layered logic”: they act frugally out of both learned habit and continued caution in response to ongoing insecurity—even when their income has increased. This complexity is now more explicitly acknowledged in the Discussion (p. 18).
  • In response to the comment about “Discrimination and Prejudice” , we have now added an interpretive paragraph linking this theme more directly to environmental practices. Specifically, we argue that experiences of marginalisation often lead to a stronger reliance on ethnic communities, which in turn reinforce low-carbon norms such as reuse, repair, and resourcefulness. This socio-spatial isolation may indirectly encourage frugal, community-based forms of sustainability that are under-recognised in mainstream frameworks.
  • Finally, we have improved the narrative flow between “Economic Shifts and New Consumption Patterns” and “Cultural Continuity”, explicitly addressing how seemingly conflicting behaviours (e.g., occasional luxury purchases vs. deep-rooted thrift) coexist. We now frame these findings as a form of adaptive hybridity: cultural identity is not static, but selectively negotiated in response to new opportunities, needs, and settings.

5. Comments: Third, I think that the added value of this study, compared to the previous research findings presented by the authors themselves in the Research background section, does not emerge in a sufficiently clear way. I would thus invite the authors to explicitly clarify their contribution to research, compared to previous literature, and to more extensively deal with the recommendations resulting from their findings. Specifically, Section 5.4 on the “Implications for policy and practice” provides interesting recommendations for future policy-making. I would suggest to further broaden them, by making them even more actionable and capable to provide direct support to policy-makers and practitioners. For instance, the small but very precise hints that are offered in the abstract (about use of WeChat, Chinese churches, and ethnic networks) could be presented a bit more extensively in the final recommendations.

Author Response:
We thank the reviewer for this valuable comment. In response, we have taken two steps:

  • We revised Section 5.4 to more clearly articulate specific, actionable recommendations for policy-makers and community organizations. This includes elaboration on the role of ethnic media (e.g., WeChat, RED, TikTok), institutions (e.g., Chinese churches, community centres), and intergenerational programs to support outreach, engagement, and communication.
  • We also added clarifications on the novel contribution of this research in comparison to prior studies. While earlier literature has explored general sustainable behavior among migrants, our study uniquely focuses on older Chinese migrants—an underrepresented demographic—and highlights the intersection of cultural memory, frugality, and structural exclusion in shaping sustainable practices. These findings contribute to discussions on inclusive climate transitions by emphasizing cultural continuity and overlooked informal behaviors as viable pathways toward sustainability.

6. Comments: Fourth, I would invite the authors to more clearly frame their theoretical background and to more explicitly use a theory to guide their analyses and discuss their results. In the “Research background” section, the authors introduce both the Theory of Planned Behaviour and the Value-Belief-Norm theory, but then in the manuscript they only refer to the latter – and in any case this is done in a quote marginal way. Should they opt for the VBN theory as a guiding reference, I would invite the authors to fully clarify its role and the way they use it. For instance, at page 17 (row 592) VBN it used to explain the practices of older migrants; in the very next rows, however, the authors deal with younger migrants, stating that their values are disconnected from their behaviours. Isn’t this “attitude-behaviour gap” phenomenon in contrast with VBN? From my point of view, it seems to challenge the use of the VBN theory, and I would appreciate the authors to provide a richer discussion about it.

Author Response:
Thank you for this important observation. We agree that the Value-Belief-Norm (VBN) theory could be more explicitly integrated throughout our analysis and discussion. In the revised manuscript, we have clarified how VBN was used both conceptually and analytically, particularly in framing the relationships between cultural values (e.g., frugality), environmental awareness, and behavioral outcomes.

To address the perceived tension between values and behaviors—especially the awareness–action gap noted among younger migrants—we added a subsection discussing the limitations of VBN in migration contexts, where behavior is often shaped not just by personal norms but also by structural constraints, cultural habitus, and historical memory.

This gap supports our incorporation of practice theory as a complementary lens, helping to explain why older migrants engage in sustainable behaviors even in the absence of explicit environmental concern. In contrast, VBN alone might not fully capture these embodied, habitual forms of action. These points are now expanded in Sections 2.1 (Research Background) and 5.1.4 (Discussion).

7. Comments: Data collection: the authors state that, compared to “sit-down” interviews, “walking” interviews allow to uncover the deeper, lived motivations behind participants’ behaviour (page 4, row 148). Advantages of walking interviews are also reported in section 5.2 “Reflections on the walking interview method”. I am not fully convinced, however, of the actual benefits of walking interviews, for the specific research goals of the authors: what are the actual gains provided by walking interviews, compared to “static” interviews, performed into the people’s homes? As the authors give high relevance to these walking interviews (starting from the title), I would appreciate the authors could elaborate a bit more on the specific benefits of walking interviews, by providing specific examples of how they enriched the findings.

Author Response:
Thank you for raising this important point. We appreciate the opportunity to clarify our rationale for using the walking interview method and its added value to our research.

In response to your query, we have revised Section 3.1 (“Research Design and Rationale”) to provide a more detailed explanation of how walking interviews contributed unique insights that would have been difficult to access through static interviews. Specifically, we found that walking through familiar environments helped participants to anchor their narratives in concrete, lived experiences. For example, participants often pointed to home-grown vegetables, reused containers, or discount store items during the interview. These spatial triggers enabled more embodied storytelling and allowed participants to reflect on their practices in real time.

Moreover, the method helped overcome initial communication barriers. Several older Chinese migrants were more reserved during seated interviews, possibly due to cultural sensitivities and perceived power dynamics. In contrast, walking interviews provided a relaxed, egalitarian setting that encouraged more open sharing. One participant, for instance, invited us into their garden to proudly showcase their sustainable lifestyle; another demonstrated how they repurposed waste materials during the walk.

8. Comments: Data collection: Table 1 reports a classification of the sample members, that is not cited in the manuscript, though I think is quite interesting: environmental pioneers, frugal practitioners, and so on. I would invite the authors to provide more information on the meaning of these classes and on the way they were identified;

Author Response:
Thank you for this valuable comment. We agree that sample size is an important consideration in qualitative research. In this study, we adopted a purposive sampling strategy aimed at maximizing diversity in migration histories, sustainability orientations, and socio-economic backgrounds. The 20 participants were selected to reflect a range of perspectives among older Chinese migrants in Greater London, including marriage migrants, former international students, labor migrants, and grandparents who migrated for family care responsibilities.

We acknowledge that the sample is not statistically representative; however, as outlined by Guest et al. (2006) and Hennink et al. (2017), qualitative studies aiming for thematic saturation typically reach sufficient depth with 12–20 participants. Our focus was on gaining rich, contextual insights rather than broad generalizations. Future studies may consider larger and comparative samples to expand on the findings.

We have now clarified this rationale in the revised manuscript under the section “Participants and Sampling Strategy.”

9. Comments: in Table 1 the authors indicate that the sample size is of 20 persons; if I am not mistaken, the place where the interviewees currently live is only mentioned towards the end of the manuscript (mostly, London). I invite the authors to provide information on the sample size and place of living directly in section 3.2 “Participant recruitment and sampling strategy.”

Author Response: Thank you for this helpful suggestion. We have revised Section 3.2 to clearly state the total sample size (20 participants) and to specify that all participants were primarily based in Greater London. We also included examples of specific boroughs—such as Kingston, Barnet, Hackney, and Islington—known for their significant Chinese communities. This change improves clarity and addresses the need to situate the sample contextually within the manuscript.

A revised sentence has been added at the beginning of Section 3.2 as follows:

“This study involved 20 older Chinese migrants aged between 55 and 85, primarily residing in Greater London. Participants were recruited from boroughs with large Chinese populations—such as Kingston, Barnet, Hackney, and Islington…”

10. Comments: Data analysis:the authors mention they performed a principal component analysis (PCA, page 6, row 224). To my knowledge, PCA is performed on quantitative data, and I am curious to know how it was used in this qualitative analysis process: I invite the authors to provide more information on the inputs and outputs of such a PCA, and on how it is performed in the framework of their qualitative analysis.

Author Response:
Thank you for raising this important point. We acknowledge that principal component analysis (PCA) is traditionally a quantitative technique. In this study, we did not perform PCA in the statistical sense. It was used in the previous questionnaire analysis.The original phrasing in the manuscript was inaccurate and may have caused confusion. We conducted a form of qualitative thematic in this study.

To avoid confusion, we have now removed the term "PCA" and clarified the nature of our analytical approach in the revised manuscript. We emphasize that the analysis was grounded in Braun and Clarke’s (2006) thematic analysis, supplemented by hierarchical thematic structuring to refine complex themes.

Reviewer 2 Report

Comments and Suggestions for Authors

Title:Low-Carbon Practices and Cultural Adaptation Among Older Chinese Migrants: Insights from Walking Interviews on Environmental Policy and Social Integration

This manuscript studies the low-carbon practices, cultural adaptation, and policy cognition of elderly Chinese immigrants in the UK in their daily environment. This research is a topic of great practical value, and the full text is written in a standardized manner. However, some problems need to be improved:

(1) The introduction section can enrich the research related to this topic, distill the differences between this study and previous ones, and highlight the contributions of this study

(2) Twenty samples were selected for this study. If feasible, it is recommended to increase the sample. Or explain the rationality and validity of the current sample selection.

(3) During the investigation process, the author designed some questions. What were the bases for these questions?

(4) Figure 1. The theme of Older Chinese Migrant Experience in Walking Interviews is not clear enough. It is suggested to replace it.

(5) This study found that there is a certain correlation between low-carbon behaviors of the elderly and culture. Can the educational level be measured?

(6) Besides the factors focused on in this study, can other factors influencing the low-carbon behaviors of the elderly also be considered for inclusion? For example, economic factors, social factors, etc.

(7) 4.4.1. External Adaptation: Language, Diet, and Consumption. Can these influencing factors be quantified? To what extent is it affected?

(8) It is suggested that the author discuss the influence of different genders, different communities, different educational levels, etc. on low-carbon behaviors.

(9) It is suggested that the author read through the entire text and make careful revisions.

Author Response

Dear Reviewer,

We sincerely thank you for your thorough and insightful feedback on our manuscript.

We deeply appreciate the time and care you devoted to reviewing our work. Your comments have been invaluable in helping us improve the clarity, depth, and scholarly contribution of the manuscript.

We have carefully considered all suggestions and have revised the manuscript accordingly. Below, we provide a point-by-point response to each comment, indicating the changes made and the corresponding sections in the revised manuscript. We hope that these revisions adequately address your concerns and strengthen the overall quality of the paper.

  1. (1) The introduction section can enrich the research related to this topic, distill the differences between this study and previous ones, and highlight the contributions of this study

Author Response:
Thank you for your constructive suggestion. In the revised Introduction, we have substantially enriched the literature review by incorporating recent studies on migrants’ environmental behaviors, cultural sustainability, and frugality . This situates our work within a broader academic context.

Furthermore, we now explicitly articulate the contributions of this study as follows:

  • Shedding light on underrepresented elderly migrant voices in sustainability discourse.
  • Highlighting cultural frugality as a form of environmental engagement, beyond formal policy participation.
  • Demonstrating the intergenerational gap in sustainability practices and values among migrants.

These points are now clearly presented in the revised Introduction and reiterated in the Discussion and Conclusion sections.

  1. (2) Twenty samples were selected for this study. If feasible, it is recommended to increase the sample. Or explain the rationality and validity of the current sample selection.

Author Response: Thank you for this thoughtful suggestion. While we acknowledge that a larger sample could potentially yield broader generalizability, our study adopted a qualitative methodology aimed at achieving depth, contextual richness, and thematic saturation rather than statistical representativeness. The sample size of 20 participants aligns with established qualitative research guidelines, which indicate that 12–20 participants are typically sufficient to reach thematic saturation in in-depth interview studies (Guest et al., 2006; Hennink et al., 2017). Participants were purposefully selected to reflect diversity in migration backgrounds, socioeconomic status, and sustainable behaviors, ensuring a wide range of perspectives. Future research may expand the sample to include more geographic regions or comparative groups. We have added a clarification in Section 3.2 to reflect this rationale.

  1. (3) During the investigation process, the author designed some questions. What were the bases for these questions?

Author Response: Thank you for raising this important point. The semi-structured interview guide was developed based on three foundations: (1) a review of prior literature on sustainability, frugality, migration, and cultural adaptation; (2) relevant theoretical frameworks such as the Value-Belief-Norm (VBN) theory, practice theory, and cultural capital theory; and (3) preliminary exploratory interviews with older Chinese migrants. This triangulated approach ensured that the questions were both theoretically grounded and culturally appropriate. We have clarified this in Section 3.3 of the manuscript.

  1. (4) Figure 1. The theme of Older Chinese Migrant Experience in Walking Interviews is not clear enough. It is suggested to replace it.

Author Response: Thank you for the helpful suggestion. In response, we have replaced the original Figure 1 with a clearer, structured table format outlining the first three levels of thematic coding (main themes, sub-themes, and sub-sub-themes). The revised version has been moved to the Appendix to improve legibility and better support the reader's understanding of the data structure.

  1. (5) This study found that there is a certain correlation between low-carbon behaviors of the elderly and culture. Can the educational level be measured?

Author Response:

Thank you for this insightful comment. In our current dataset, participants’ educational levels were not explicitly collected as a formal demographic variable, primarily due to the exploratory and narrative-driven nature of the walking interview method. However, through participants’ self-disclosures during interviews, we observed a wide range of informal educational experiences—from primary school education to university-level backgrounds.

That said, we agree that formal education may influence awareness and framing of environmental behaviors, especially in how individuals articulate sustainability using policy or scientific language. However, in our study, low-carbon behaviors such as thrift, reuse, and waste avoidance were consistently practiced across different perceived educational levels, suggesting that such behaviors were more strongly rooted in cultural memory and personal history than in formal environmental education.

  1. (6) Besides the factors focused on in this study, can other factors influencing the low-carbon behaviors of the elderly also be considered for inclusion? For example, economic factors, social factors, etc.

Author Response:

Thank you for this thoughtful suggestion. We agree that cultural factors alone do not fully explain the complexity of low-carbon behaviors among older migrants. Indeed, economic and social factors emerged throughout the interviews, though they were not always treated as standalone analytical categories. For instance, participants frequently cited rising living costs and utility bills as motivations for energy conservation, indicating that financial concerns interact with culturally embedded frugality.

Moreover, social isolation or reliance on ethnic networks also shaped behavioral patterns, particularly regarding reuse and community-based resource sharing. We have now elaborated on these interrelated economic and social dimensions in the Discussion section and emphasized their relevance for future research.

  1. (7) 4.4.1. External Adaptation: Language, Diet, and Consumption. Can these influencing factors be quantified? To what extent is it affected?

Author Response:

Thank you for this insightful question. We agree that elements such as language proficiency, dietary transitions, and consumption habits represent key dimensions of cultural adaptation and could be measured more systematically. While this study utilized a qualitative, walking-interview methodology to explore lived experiences in context, we recognize that future research could incorporate quantitative tools to assess the extent and variability of these adaptation processes.

  1. (8) It is suggested that the author discuss the influence of different genders, different communities, different educational levels, etc. on low-carbon behaviors.

Author Response:

Thank you very much for your constructive comment.  We fully agree that factors such as gender, community context, and educational attainment are crucial in shaping low-carbon behavior and warrant further investigation.

In our current study, we prioritized cultural values, migration experience, and generational perspectives to understand low-carbon practices among older Chinese migrants.  However, we acknowledge that gender roles and educational backgrounds may intersect with these dimensions in meaningful ways.  For example, preliminary patterns suggest that older women in our sample were more likely to engage in home-based frugal practices, such as food preservation and reuse of materials, while men discussed more structural aspects of housing or energy use.  Educational background also influenced how participants interpreted or framed their actions—for instance, those with higher education levels more often referenced scientific or policy-related terms.

Reviewer 3 Report

Comments and Suggestions for Authors

Sustainability research has largely overlooked the implications of transitions like decarbonization for the lives of people who differ from a country or society's mainstream cultural groups. This study offers a valuable and insightful perspective by shedding light on this important yet neglected issue. Furthermore, the analysis of the Walking Interviews is very thorough and convincing. I strongly encourage you to further refine and publish this highly significant research.

While I greatly appreciate this research, I have identified several areas for improvement that I believe would enhance it further.

  1. Introduction: This section is not very strong. It bears an unfavorable resemblance to the often overly simplistic research common in Sustainability, Cleaner Production, and Sustainable Consumption and Production, which suggests that "changing citizens' behavior will save the world." In reality, the brilliance of this research lies in its observation of how Chinese immigrants form their new living practices while being influenced by their experiences before coming to the UK and the differences in institutions, living environments, relationships with other groups, and newly available information like social media. This leads to the accurate observations that a) this is not uniform among all Chinese immigrants but varies across generations, and b) while there are aspects that are incompatible with the mainstream low-carbon living discourse, there are also similarities in practical terms. This could be a significant critique of the linear thinking often found in national and local government policies, which assume that "providing information to citizens, shaping their sustainable values, and changing their behavior" implicitly targets only those who are already immersed in consumer culture and are close to the cultural, social, and economic mainstream within the country or society (or at least not marginalized). It also serves as a critique of the Theory of Planned Behavior (TPB) and Environmental Citizenship theories. By analyzing how such cultural transitions, frictions, and adaptations differ across cultural groups, generations, and genders, and how they involve different burdens and opportunities, we can elevate the discussion of inclusive and just transitions to a significantly more substantial level. If I may, I would suggest comprehensively revising the initial section with this perspective in mind.

    By the way, regarding the expression of a 40-70% reduction potential in Line 35, when this figure appears in reports such as the IPCC 2022 report, it is used in the context of reductions through "demand-side mitigation" and not specifically for the reduction potential through consumer behavior. I apologize if I am mistaken.

  2. Research Background: This section is very interesting, but in relation to the points mentioned above, there are areas that might benefit from revision. 2.1. The methodology and the identification of the research problem in this study severely critique the assumptions of TPB and seem closer to discussions of practice theory and cultural capital. The author team should reconsider the theoretical explanations. 2.2. Frugality & 2.4. Migrants' Sustainability Practice These descriptions are very interesting. It might be better to structure the rationale for choosing "migrants' decarbonization behavior" as the research subject based on this content.

  3. Results: The text in Figure 1 is too small and difficult to read. Could you explore any ways to improve this? The content itself is very interesting. (This is not a comment directly on the paper, so you don't need to reflect it during revisions.) Interviews have an aspect of co-constructing narratives through the relationship between the speaker and the listener who are present at that specific time and place. I understand that using software like Nvivo to extract key categories from transcribed text is a valuable method. However, I am concerned that the increasing sophistication of techniques to ensure objectivity might be undermining the significance of interviews as "context-specific co-constructions of narratives."

  4. Discussion: Along with the Introduction, this section was also disappointing. Given the insightful observations in the Results section, I believe a deeper analysis could be developed. Could you re-examine the generational differences (past experiences, social integration and adaptation, the meaning of frugality, information tools used, perspectives on mainstream policies) and the living situation in the UK (well-developed social security, shopping opportunities, proximity to consumer culture)? Based on this, could you analyze the unique yet complex meanings and future prospects of their "decarbonization behavior" and "frugality," which are not monolithic? Furthermore, could you then reconsider how the mainstream discourse on "low-carbon lifestyles" and sustainable consumption has been non-inclusive and has ignored the complexities of cultural formation?

I hope these comments are helpful in further improving your research.

Author Response

Dear Reviewer,

We sincerely thank you for your thorough and insightful feedback on our manuscript.

We deeply appreciate the time and care you devoted to reviewing our work. Your comments have been invaluable in helping us improve the clarity, depth, and scholarly contribution of the manuscript.

We have carefully considered all suggestions and have revised the manuscript accordingly. Below, we provide a point-by-point response to each comment, indicating the changes made and the corresponding sections in the revised manuscript. We hope that these revisions adequately address your concerns and strengthen the overall quality of the paper.

Comments:

Introduction: This section is not very strong. It bears an unfavorable resemblance to the often overly simplistic research common in Sustainability, Cleaner Production, and Sustainable Consumption and Production, which suggests that "changing citizens' behavior will save the world." In reality, the brilliance of this research lies in its observation of how Chinese immigrants form their new living practices while being influenced by their experiences before coming to the UK and the differences in institutions, living environments, relationships with other groups, and newly available information like social media. This leads to the accurate observations that a) this is not uniform among all Chinese immigrants but varies across generations, and b) while there are aspects that are incompatible with the mainstream low-carbon living discourse, there are also similarities in practical terms. This could be a significant critique of the linear thinking often found in national and local government policies, which assume that "providing information to citizens, shaping their sustainable values, and changing their behavior" implicitly targets only those who are already immersed in consumer culture and are close to the cultural, social, and economic mainstream within the country or society (or at least not marginalized). It also serves as a critique of the Theory of Planned Behavior (TPB) and Environmental Citizenship theories. By analyzing how such cultural transitions, frictions, and adaptations differ across cultural groups, generations, and genders, and how they involve different burdens and opportunities, we can elevate the discussion of inclusive and just transitions to a significantly more substantial level. If I may, I would suggest comprehensively revising the initial section with this perspective in mind.

Response: We sincerely thank the reviewer for this insightful and constructive critique. In response, we have substantially revised the Introduction section to better reflect the complexity and richness of our study.

Specifically, we have:

  • Avoided linear models of behavior change, and emphasized that sustainability is not solely a matter of informed individual choice, but is shaped by culture, migration experience, and everyday adaptation.
  • Expanded the theoretical framing by incorporating not only the Value-Belief-Norm (VBN) theory, but also practice theory and cultural capital theory, as suggested. These frameworks allow us to analyze sustainability practices as embedded in historical experiences, routines, and values.
  • Clarified our critical contribution to dominant sustainability discourse by highlighting how older Chinese migrants practice sustainability outside formal environmental frameworks—thus offering an inclusive perspectiveon low-carbon transitions.
  • Foregrounded cultural frictions and generational tensions, as well as the implications for more just and inclusive environmental policy.

Comments:

By the way, regarding the expression of a 40-70% reduction potential in Line 35, when this figure appears in reports such as the IPCC 2022 report, it is used in the context of reductions through "demand-side mitigation" and not specifically for the reduction potential through consumer behavior. I apologize if I am mistaken.

Response: Thank you for pointing out the imprecise parts. We changed this sentence to : Recent estimates suggest that demand-side mitigation strategies—including changes in infrastructure, technology, and individual behaviour—could reduce global greenhouse gas emissions by 40–70% by 2050 compared to current policy projections \cite{IPCC2022}.

Comments:

Research Background: This section is very interesting, but in relation to the points mentioned above, there are areas that might benefit from revision. 2.1. The methodology and the identification of the research problem in this study severely critique the assumptions of TPB and seem closer to discussions of practice theory and cultural capital. The author team should reconsider the theoretical explanations. 2.2. Frugality & 2.4. Migrants' Sustainability Practice These descriptions are very interesting. It might be better to structure the rationale for choosing "migrants' decarbonization behavior" as the research subject based on this content.

Response: Thank you very much for this insightful and constructive suggestion. We fully agree that our study critiques the linear assumptions of TPB and aligns more closely with practice theory and the notion of cultural capital. In response, we have revised the Theoretical Framework section to explicitly reflect this orientation.

In addition to using the Value-Belief-Norm (VBN) theory to understand moral drivers of sustainable behavior, we have now included practice theory (Shove et al., 2012) and Bourdieu’s concept of cultural capital (Bourdieu, 1986) to better capture the embodied, routinized, and historically grounded aspects of older Chinese migrants’ low-carbon actions. These perspectives help explain how thrift and resource-conscious practices arise not from abstract pro-environmental beliefs but from inherited cultural norms and lived histories of scarcity.

Specifically, we added the following paragraph to the revised manuscript:

“While we draw on the Value-Belief-Norm (VBN) theory to explain how moral and normative values influence low-carbon behaviors, we also incorporate perspectives from practice theory and cultural capital theory. These approaches shift the analytical focus from individual attitudes and intentions to embodied routines, material arrangements, and inherited social values. In this view, frugality does not emerge in response to climate discourse but as a durable practice rooted in Confucian ethics, postwar scarcity, and financial prudence. This helps to illuminate how culturally embedded habits, rather than environmental ideology—can shape sustainable actions in migrant communities.”

Comments:

Results: The text in Figure 1 is too small and difficult to read. Could you explore any ways to improve this? The content itself is very interesting. (This is not a comment directly on the paper, so you don't need to reflect it during revisions.) Interviews have an aspect of co-constructing narratives through the relationship between the speaker and the listener who are present at that specific time and place. I understand that using software like Nvivo to extract key categories from transcribed text is a valuable method. However, I am concerned that the increasing sophistication of techniques to ensure objectivity might be undermining the significance of interviews as "context-specific co-constructions of narratives."

Response: Thank you very much for this thoughtful and important reflection. In response to the readability issue, we have replaced the original Figure 1 with a clean, well-labeled thematic table, now presented in the Appendix (Appendix A). This revised format improves clarity and allows readers to engage more easily with the layered thematic structure of our findings.

Regarding your epistemological point, we wholeheartedly agree that interviews are not merely repositories of data but are situated, dialogic, and co-constructed narratives. Our walking interviews, in particular, were designed to foreground this relational and contextual dimension. Although we use NVivo as an auxiliary tool to help manage and group emergency code, our topic analysis is explanatory, and the author recorded the interaction details with the respondents through diaries. In the future, the author will consider whether there are better analysis methods and tools to better maintain the narrative nature of the situation.

Comments:

Discussion: Along with the Introduction, this section was also disappointing. Given the insightful observations in the Results section, I believe a deeper analysis could be developed. Could you re-examine the generational differences (past experiences, social integration and adaptation, the meaning of frugality, information tools used, perspectives on mainstream policies) and the living situation in the UK (well-developed social security, shopping opportunities, proximity to consumer culture)? Based on this, could you analyze the unique yet complex meanings and future prospects of their "decarbonization behavior" and "frugality," which are not monolithic? Furthermore, could you then reconsider how the mainstream discourse on "low-carbon lifestyles" and sustainable consumption has been non-inclusive and has ignored the complexities of cultural formation?

Response: Thank you very much for this thoughtful and constructive comment. In response, we have substantially revised the Discussion section to deepen the analysis in several key ways:

  • Generational Differences: We now expand on intergenerational contrasts by integrating participants’ reflections on how younger migrants, though more environmentally aware (via education and media exposure), may not engage in consistent sustainable practices. This perceived awareness–action gap is used to challenge the assumptions of linear models like VBN and highlight the need for more context-sensitive theories. We clarify that these insights are based on older migrants’ perceptions, not direct interviews with younger participants (see revised Section 5.1.4).
  • Living Conditions and Cultural Adaptation: We now explore how UK living conditions—such as access to consumer goods, insulation programs, or social security—intersect with older migrants’ previous experiences in China. This contrast helps explain both persistence and transformation in behavior, and how new environments reshape but do not fully override long-standing cultural logics of frugality (see Section 5.1.1 and 5.1.2).
  • Complexities of Frugality and Decarbonization: We discuss how frugality, though rooted in historical scarcity, has taken on layered meanings in the migration context—sometimes as a choice, sometimes as a necessity, and sometimes as resistance to wasteful norms. We also reflect on how these behaviors are interpreted differently across generations and cultural contexts (Section 5.1.1–5.1.2).
  • Critique of Mainstream Sustainability Narratives: We added a subsection titled “Rethinking Inclusion in Environmental Narratives”, which critiques the dominant policy focus on consumer choice and technological solutions. We argue that such frameworks often exclude the “quiet sustainability” practiced by older, marginalized communities. This builds on the reviewer’s insight regarding the exclusionary nature of mainstream discourse and ties it to broader concerns about environmental justice (Section 5.1.5).
  • Integration of Cultural, Economic, and Social Drivers: A new subsection (5.1.6) explicitly analyzes how economic pressures (e.g., utility bills), cultural memory, and social exclusion interact to shape sustainable practices—responding to the suggestion to go beyond cultural framing alone.

Round 2

Reviewer 2 Report

Comments and Suggestions for Authors

After receiving the revised, the overall quality has been improved, it is recommended that the author read through the full text and carefully check.

Reviewer 3 Report

Comments and Suggestions for Authors

The authors have successfully responded sincerely to the review comments and repositioned this research from a simple study of Behaviour Change to a more fruitful area focusing on the social adaptation, practices, and cultural transmission and change of minorities in a changing society. As the authors argue, research on Transition, particularly that related to Behaviour Change, tends to focus solely on the socially, culturally, and financially privileged majority in society. In this context, the choice of Chinese immigrants to the UK as the target of this study is appropriate. Furthermore, the advantages of adopting the Walking Interview method are very clearly stated, and looking at the interview results in the Results section, the effectiveness of this method is evident. Overall, the paper has been revised into a valuable research paper that fully utilizes the strengths of the original interview data.